# Characterisation of the Semliki Forest Virus-host cell interactome reveals the viral capsid protein as an inhibitor of nonsense-mediated mRNA decay

Lara Contu [1,2], Giuseppe Balistreri [3,4], Michal Domanski [1], Anne-Christine Uldry [5], Oliver Mühlemann [1] *

**1** Department of Chemistry and Biochemistry, University of Bern, Bern, Switzerland, **2** Graduate School for Cellular and Biomedical Sciences, University of Bern, Bern, Switzerland, **3** Faculty of Biological and Environmental Sciences, University of Helsinki, Helsinki, Finland, **4** Queensland Brain Institute, The University of Queensland, Brisbane, Australia, **5** Proteomics & Mass Spectrometry Core Facility, Department for BioMedical Research, University of Bern, Bern, Switzerland

* oliver.muehlemann@dcb.unibe.ch

**Data Availability Statement:** The data supporting the findings of this study are available within the paper and its supplementary files (S1 Table, S2

## Abstract

The positive-sense, single-stranded RNA alphaviruses pose a potential epidemic threat. Understanding the complex interactions between the viral and the host cell proteins is crucial for elucidating the mechanisms underlying successful virus replication strategies and for developing specific antiviral interventions. Here we present the first comprehensive protein-protein interaction map between the proteins of Semliki Forest Virus (SFV), a mosquito-borne member of the alphaviruses, and host cell proteins. Among the many identified cellular interactors of SFV proteins, the enrichment of factors involved in translation and non-sense-mediated mRNA decay (NMD) was striking, reflecting the virus' hijacking of the translation machinery and indicating viral countermeasures for escaping NMD by inhibiting NMD at later time points during the infectious cycle. In addition to observing a general inhibition of NMD about 4 hours post infection, we also demonstrate that transient expression of the SFV capsid protein is sufficient to inhibit NMD in cells, suggesting that the massive production of capsid protein during the SFV reproduction cycle is responsible for NMD inhibition.

## Author summary

To take over control of the host cell and ensure its own replication, viral proteins interact with a plethora of host cell proteins. Elucidating these viral–host cell protein interactions is therefore key for understanding the mechanisms that a virus applies to successfully hijack the host cell. This study provides the first comprehensive protein–protein interaction map between the proteins of Semliki Forest Virus (SFV), a positive-strand, single-stranded RNA virus of the alphavirus family. While we previously discovered that the host cell recognises and degrades the incoming viral genomic RNA by a

Table, S3 Table). The mass spectrometry proteomics data have, in addition, been deposited to the ProteomeXchange Consortium via the PRIDE database identifier PXD022036 (https://www.ebi.ac.uk/pride/).

**Funding:** This work was financially supported by the NCCR RNA & Disease funded by the Swiss National Science Foundation (SNSF) grants 51NF40-182880, by SNSF grants 31003A-162986 and 310030B-182831, and by the canton of Bern to OM. The funders had no role in the study design, data collection and analysis, decision to publish, or preparation of the manuscript.

**Competing interests:** The authors have declared that no competing interests exist.

cellular quality control system called Nonsense-Mediated mRNA Decay (NMD), our interactome study now led to uncovering of the other side of this arms race between SFV and the infected cells: We show in this study that the viral capsid protein has the capacity to inhibit NMD.

## Introduction

As we live through the current SARS-COV2 pandemic, the world is reminded of the unpredictable nature of viral epidemics and the importance of studying potential emerging viral threats. Recent studies present valid arguments for the worldwide epidemic threat of alphaviruses (among other arboviruses) that currently circulate endemically in particular regions [1,2]. The outbreak potential of alphaviruses has already been showcased by the two worldwide epidemics caused by Chikungunya virus (CHIKV) that affected more than 8 million people in over 50 countries and could be attributed to a single point mutation leading to a 100-fold increase in infectious virus in the salivary glands of urban mosquitoes [1,2]. This demonstrates that small genetic alterations can cause dramatic changes in human transmissibility and infection. Semliki Forest Virus (SFV) is closely related to CHIKV, both evolutionarily grouped within the Semliki Forest (SF) clade of the Old World alphaviruses (Family: *Togaviridae*) [3]. SFV causes lethal encephalitis in mice [4]. Though mostly associated with mild febrile illness or asymptomaticity in humans, SFV is endemic to African regions [1] and a handful of studies indicate serious disease relevant symptoms associated with SFV in humans, including encephalitis, myalgia and arthralgia [5–8].

SFV is a small (~70 nm in diameter), enveloped virus comprising a nucleocapsid core made up of 240 copies of capsid protein that surrounds its positive-sense single-stranded RNA genome (~11.8 kb). The genome contains a 5′ cap (N7mGppp) and poly(A) tail and is organised into two distinct open reading frames (ORFs). The first ORF encodes the non-structural proteins (nsP1, nsP2, nsP3 and nsP4) (Fig 1A), which are translated as one polyprotein (P1234) immediately upon exposure of the viral mRNA-genome to the cytoplasm [9–12]. The polyprotein is then proteolytically cleaved by the protease activity of nsP2 to yield functional viral replicase complexes [13]. The first protein to be cleaved from the polyprotein is nsP4, comprising RNA-dependent RNA polymerase activity. The resulting P123 polyprotein in complex with nsP4 forms the viral replication complex (RC), responsible for synthesizing minus strand template RNA from the genomic viral (v)-RNA early during infection [9]. The ensuing double-stranded vRNA intermediates can trigger the activation of Protein Kinase double-stranded RNA-dependent (PKR), resulting in phosphorylation of the α-subunit of the eukaryotic translation Initiation Factor 2 (eIF2) and thus causing a decrease in global translation of host cell messenger RNAs (mRNAs) [10,14,15]. As proteolytic cleavage of P123 by nsP2 progresses, individual nsPs form new viral RCs of altered composition, resulting in a shift from synthesis of the minus strand template, to synthesis of new viral genomes and viral subgenomic RNA (sgRNA) from the 26S promoter (Fig 1A) [9,16]. Alphavirus replication occurs in membrane invaginations called 'spherules', where high concentrations of RCs are present [9,11]. Binding of host cell proteins to RCs has been reported, though the abundances and the functions thereof are still not fully understood [11,17,18]. In addition, individual SFV proteins localise independently of the RC to perform functions separate from viral replication [9,11,19,20]. One example is the nsP2 protein, which translocates to the nucleus [21] and has been shown to suppress host cell transcription through induction of polyubiquitination

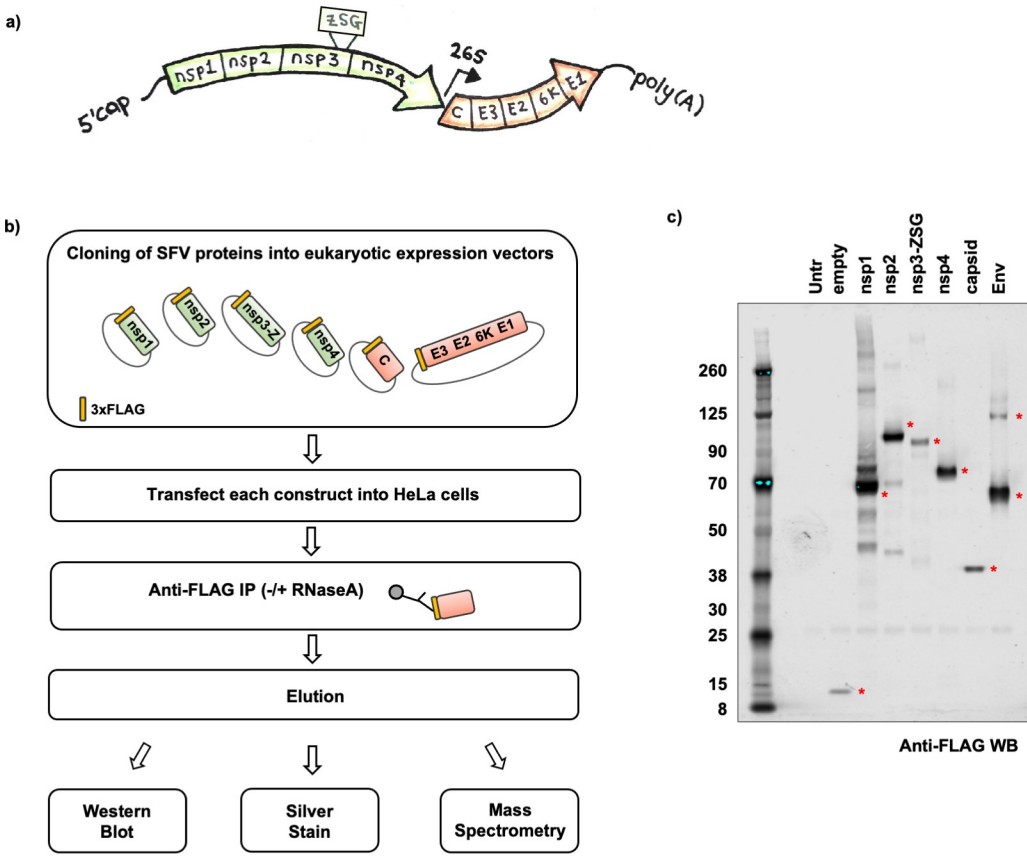

**Fig 1. Strategy for creation of Semliki Forest Virus (SFV)–host protein-protein interaction map. a,** Schematic illustration of the genomic organisation of SFV. The first ORF encoding the non-structural proteins (nsp1, nsp2, nsp3, nsp4) is highlighted in green. The ZSG tag inserted within the nsp3 protein is also depicted. The second ORF encoding the structural proteins (capsid, E3,E2,6K,E1) is highlighted in orange. Other viral features depicted include the 5′ cap, poly(A) tail, as well as the position of the 26S subgenomic viral promoter. **b,** Flowchart outlining the experimental approach to transiently express N-terminally 3xFLAG-tagged (yellow rectangle) SFV proteins in mammalian cells in order to construct a SFV-host protein-protein interactome. Nsp3-Z refers to the nsp3 protein with the ZSG tag, C refers to the capsid, and E3 E2 6K E1 refer to the envelope proteins (Env) that were expressed as one polyprotein. **c,** Anti-FLAG western blot of SFV proteins after transient transfection and affinity purification from HeLa cells (without RNase A treatment). Red asterisks indicate 3xFLAG tagged SFV proteins at their expected sizes. Untransfected cells (Untr) and cells transfected with a plasmid encoding only the 3xFLAG tag with no additional coding region (empty) were included as controls. The expected sizes of the 3xFLAG-tagged proteins were: empty ~8kDa; nsp1 ~63kDa; nsp2 ~92kDa; nsp3-Z ~82kDa; nsp4 ~72kDa; capsid ~33kDa and Env (polyprotein ~111kDa, cleavage intermediates ~57kDa/ ~64kDa). The affinity purifications were conducted in triplicate (± RNase A treatment), and eluates analysed by mass spectrometry.

followed by rapid degradation of Rpb1, a catalytic subunit of the RNA polymerase II complex [22].

The second ORF of SFV encodes the structural proteins (Fig 1A), which are translated as a single polyprotein (C-E3-E2-6K-E1) from the sgRNA later during infection [23]. SgRNA translation occurs despite phosphorylation-induced inactivation of eIF2α [10,17,24–27]. Once translated, the capsid (C) autoproteolytically cleaves itself from the growing polypeptide chain, while the remaining polyprotein is translocated into the Endoplasmic Reticulum (ER) lumen and processed by cellular enzymes into the glycoproteins, precursor E2 (pE2), E1 and the small membrane protein, 6K. These move through vesicles in the secretory pathway to the plasma membrane (PM), during which pE2 is further cleaved into E3 and E2 glycoproteins [23]. Nucleocapsids are formed through binding of capsid proteins to the vRNA [17].

Encapsidated viral genomes interact with the cytoplasmic domains of the glycoproteins exposed at the inner side of the PM from where they bud as mature, enveloped, infectious viral particles [28].

Here, we investigated the virus-host protein interactome of SFV. A greater understanding of the repertoire of host proteins that may be exploited by viruses is a vital first step toward developing antiviral strategies aimed at targeting or interfering with interactions that may be critical for the infection. While previous studies have reported host interactors of SFV from isolations of RCs from lysosome fractions, as well as affinity purifications and localisation studies of recombinant virus in which nsP3 was tagged with the fluorescent protein ZsGreen (ZsG) [18,29–31], there are so far no SFV studies that assess the complete set of viral-host protein-protein interactors (PPI). Using affinity purifications followed by high-throughput quantitative mass spectrometry, we identified host protein interactors of individual SFV proteins in human cells. In addition, using a genome-wide siRNA screen we assigned pro or antiviral functions to some of the identified SFV interactors. Gene ontology (GO) enrichment analyses of protein complexes that could form between the identified host interactors revealed highly significant GO terms related to translation and Nonsense-mediated mRNA Decay (NMD). NMD is known to restrict infection of alphaviruses, but whether and how the virus counteracts this cellular intrinsic defence is still not clear [32]. Here we show that during the course of infection SFV suppresses NMD. We present evidence that the capsid protein of SFV is sufficient to suppress NMD independently of translation inhibition.

## Results

As obligatory parasites, all viruses exploit the host cell to favour their own replication. In turn, cells have evolved mechanisms to protect against viral infections. To gain insight into the repertoire of host proteins that could be exploited by SFV, we systematically mapped the interactions between the individual SFV proteins, nsP1-4, C, and the envelope polyprotein, Env (which includes E3, E2, 6K and E1), and the host cell proteome using high-throughput quantitative mass spectrometry (Fig 1A). The SFV proteins were N-terminally tagged with 3xFLAG and transiently expressed in HeLa cells, a cell type susceptible to infection by SFV [33]. The proteins were then affinity purified from the respective lysates using anti-FLAG antibodies, with and without treatment with RNase A to distinguish RNA-mediated from protein-mediated interactions (Fig 1B). Western blot analysis of the eluates from each anti-FLAG affinity purification revealed the successful pulldown of all six transiently expressed SFV proteins (Fig 1C).

The protein compositions of the eluates, from three biological replicates of each affinity purified SFV protein, were analysed by quantitative mass spectrometry (Figs 2A and S1). Significant interactors (see Material and Methods) (Fig 2B, purple circles and S1 Table) were further filtered by abundance, such that proteins whose abundance made up at least 0.5% of the relevant SFV bait protein were retained (Fig 2B, black crosses and S1 Table). In the case of the nsP3 bait (here fused with ZsG as in the recombinant virus, nsP3-Z), which was very lowly abundant in the sample as it proved difficult to elute from the beads (S2 Fig), we retained proteins whose abundance made up at least 5% of the bait (S1 Table). The heat map in Fig 2C summarises the most abundant significant interactors of each SFV protein in the–RNase A samples, with their corresponding abundance in the +RNase A samples alongside them. Many of the host interactors identified in the–RNase A sample were lost upon treatment with RNase A, indicating that these interactions were likely mediated by RNA. This was clear for many nsP2, nsP3-Z and capsid interactors, where the heat map (Fig 2C) corroborated the observations in the analytical silver stain gel (Fig 2A). In both the heat map and the gel, we noted

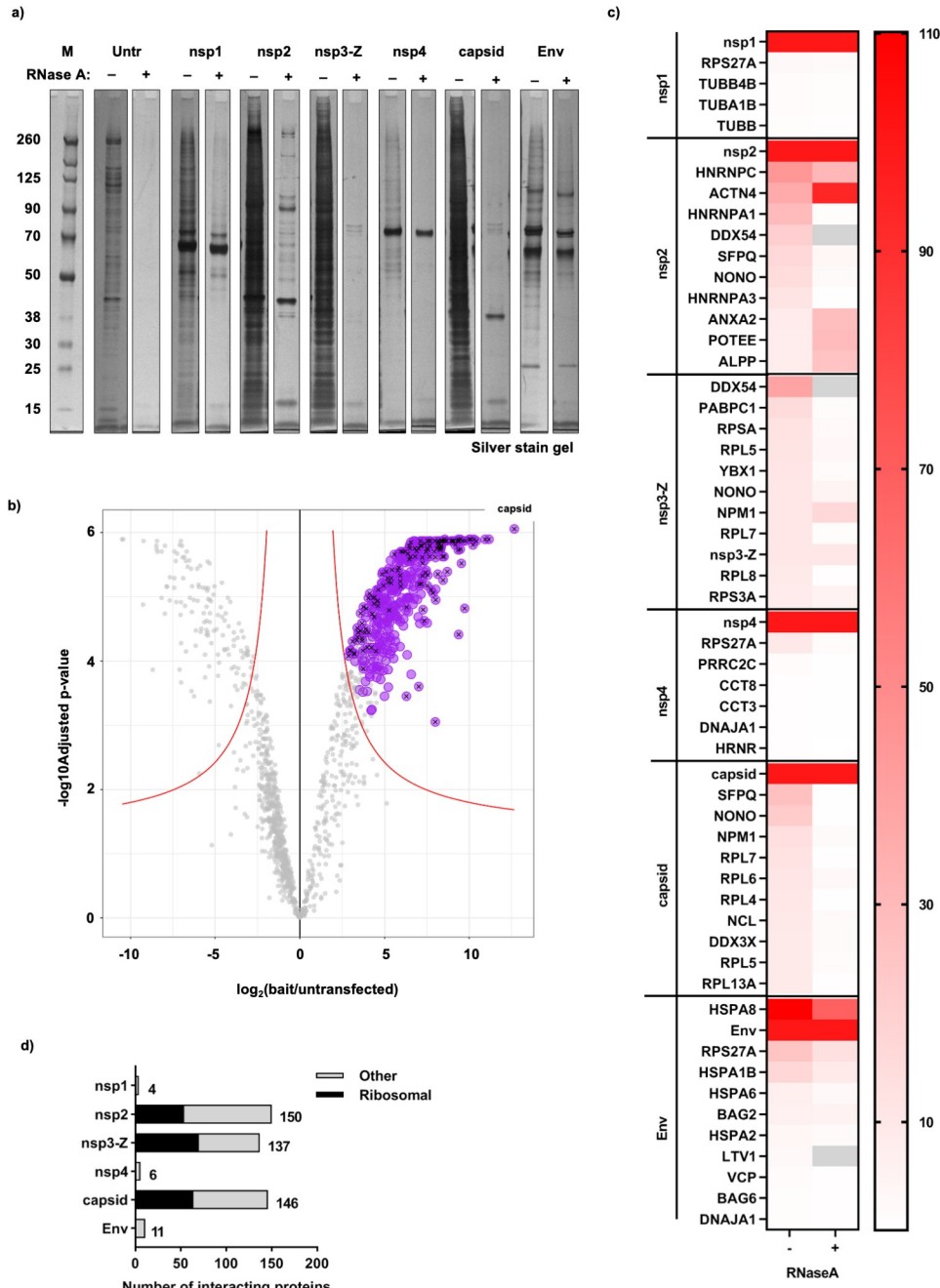

**Fig 2. Summary of host protein interactors of individual SFV proteins, revealed by mass spectrometry analysis. a,** Analytical silver stained gel showing the SFV affinity purification eluates that were analysed by mass spectrometry (for both ± RNase A treated samples). **b,** Volcano Plot showing significantly enriched proteins in the SFV capsid protein sample (–RNase A) compared to the untransfected control. The capsid sample was selected to display an example of the mass spectrometry analysis that was performed for all SFV baits: iTop3 values were used to perform Differential Expression tests (by applying the Empirical Bayes tests) for each test sample (SFV bait) compared to the untransfected control sample. The log2FC and–log10(adjusted p-value) values from each analysis from three biological replicates were plotted. A significance curve (red) was calculated based on a minimal log2FC of 1 and a maximum adjusted p-value of 0.05. All proteins that were (i) significantly enriched in the SFV capsid sample compared to the control, (ii) persistently significant through the imputation cycles, and (iii) identified as 'true' interactors by SAINT analysis (FDR< = 0.05) are depicted in purple. Black crosses indicate proteins with an abundance threshold of at least 0.5% of the capsid bait. The position of the SFV capsid bait protein in the volcano plot is depicted. **c,** Using the list of–RNase A interactors, a threshold of abundance of at least 0.5% of the bait protein (in the case of nsp3-Z, at least 5% of the bait protein) was applied. A heat map summarising either the top10 most abundant interactors (or all if there were fewer

than 10 interactors identified) for each SFV bait protein without RNase A treatment (–) is shown. The corresponding abundance as % of bait of the interactors that were statistically significantly enriched when treated with RNase A (+) is also shown in the heat map. Grey blocks indicate proteins that did not appear or were not statistically significantly enriched in the +RNase A samples. Note that due to the low abundance of nsp3-Z, the abundance as % of bait of nsp3-Z and all its interactors are presented as a factor of 10 less than what was calculated (for better visual representation of the heat map as a whole). For all values and the complete list of interactors, see S1 Table. **d,** Bar graph summary of the number of interactors collected for each bait, with the fraction of ribosomal protein interactors depicted in black, and all other host interactors depicted in grey.

proteins in the nsP2 eluate that were enriched in the +RNase A sample compared to the–RNase A sample. Also reflected in the heat map were proteins observed in the gel that were more than or as abundant as nsP2 (~92 kDa) (Fig 2A and 2C, +RNase A samples). The quantified lists obtained to create the heat map were overall consistent with the patterns observed on the silver-stained gels. Since many SFV-host protein interactions were dependent on RNA, we chose to focus on the lists of interactors from the–RNase A datasets going forward. A summary of these revealed that a large fraction of the interactions for nsP2, nsP3-Z and capsid consisted of ribosomal proteins (Fig 2D).

These stringent lists of interactors are displayed as SFV-host interactome networks (Figs 3A and S3). Affinity purified SFV proteins are displayed as black circles, while host cell proteins are displayed as smaller, colour-coded circles. Host proteins that were identified as unique interactors to one of the SFV proteins are connected to the relevant host protein with a grey line. Many of the host proteins were identified as interactors to more than one of the SFV proteins. For simplicity, these non-unique interactors are grouped into grey boxes with grey lines connecting the whole group of proteins to the SFV proteins for which they were identified as interactors (Fig 3A). Considering this overlap, the total number of host proteins that were identified as interactors was 251 (Figs 3A and S3), 77 of which were ribosomal proteins and are shown separately (S3 Fig). Host proteins displayed in the networks were manually curated and categorised into colour-coded groups based on descriptions gathered from both Gene Ontology (GO) and STRING analyses (Fig 3A). Interactions that stood out included subunits of the chaperonin-containing t-complex polypeptide 1 (CCT complex) (pink) that interacted with both nsP2 and nsP4, a number of cytoskeletal proteins or proteins involved in cytoskeletal signalling (grey) interacting with nsP2 and nsP1 (tubulins), ER chaperones (pink) bound uniquely to Env, and a large number of RNA binding proteins (violet) interacting with nsP2, nsP3-Z and capsid. In addition, a striking presence of rRNA processing / ribosome biogenesis factors emerged as interactors, many of which were found bound uniquely to the capsid (dark pink) (Fig 3A). Previously reported human protein-protein interactions were analysed by STRING and additionally displayed on the networks (pink dashed lines). The dense network of edges (pink dashed lines) that emerged among the rRNA processing/ribosome biogenesis factors (dark pink) reflects the known protein-protein interactions that have been reported between them (Fig 3A). This indicates that the capsid protein (and to a lesser extent nsP2/nsP3-Z) may interact with a complex of proteins involved in rRNA processing and/or ribosome biogenesis.

We next set out to determine whether the identified host proteins have pro- or antiviral effects in the context of infection. To systematically address this question, we used a genome-wide fluorescence microscopy-based siRNA screen [33] to identify host proteins that affected SFV replication (S2 Table). In this screen, a recombinant SFV expressing the ZsGreen fluorescent protein fused in frame with nsP3 (SFV-ZSG) was used, allowing for the quantification of the fraction of infected cells by automated fluorescence imaging. Thus, in the infection assays, the sequences of the viral proteins are the same as those used in our interactome experiment.

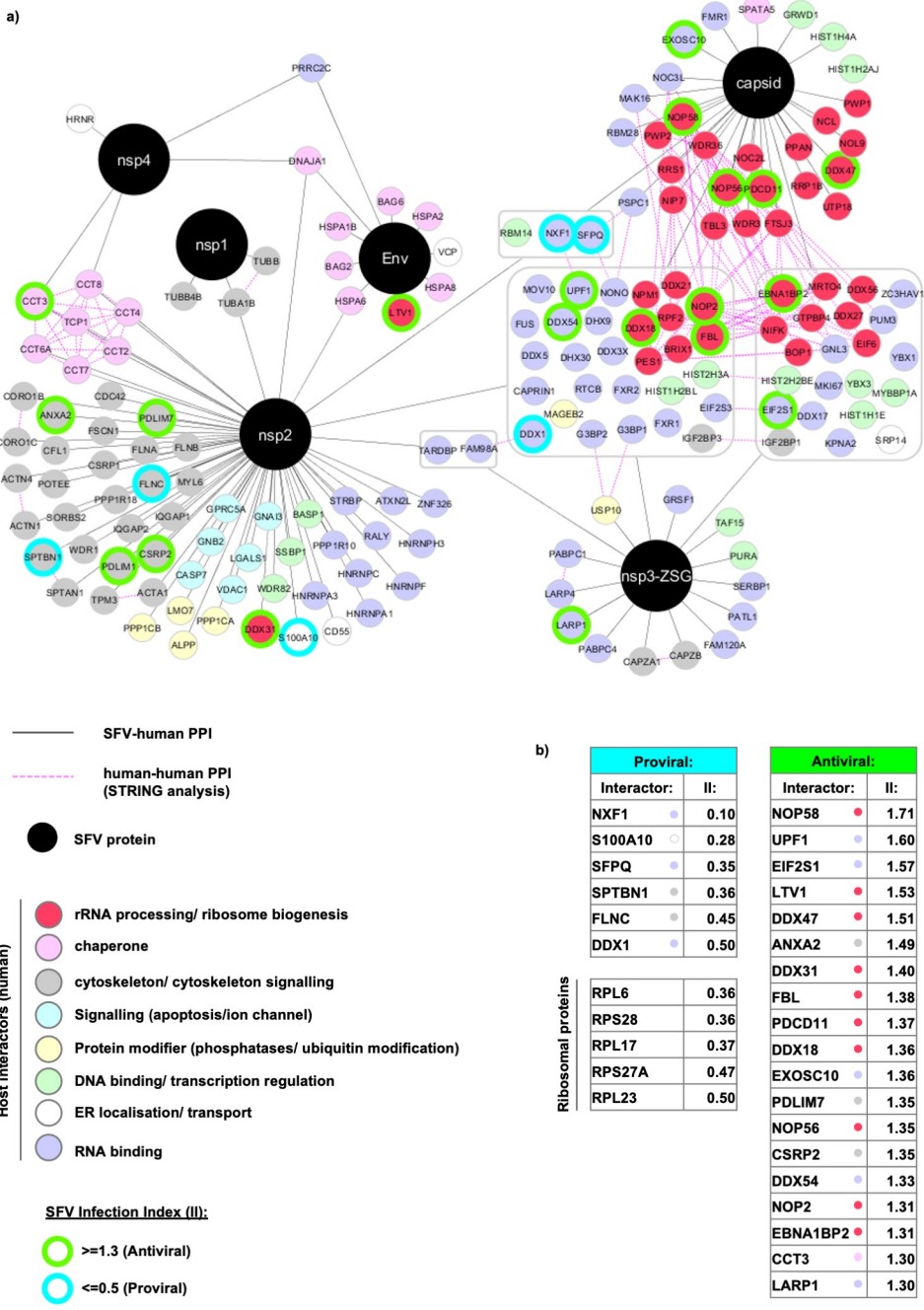

**Fig 3. Network visualisation of the SFV-host protein interactome. a,** In total, 174 host proteins are displayed. SFV proteins are depicted in black circles and host proteins are displayed in smaller colour-coded circles according to the key. SFV-host protein-protein interactions (PPI) are depicted with solid grey lines. Grey boxes reflect host proteins identified as interactors to more than one SFV protein. In these cases, the solid grey lines connect the grouped set of host proteins to the SFV proteins for which they were identified as interactors. Host-host PPI ascertained through STRING analysis are depicted with dashed pink lines. The host interactors were collected based on three independent biological replicates. For simplicity, ribosomal proteins (77) are shown separately in S3 Fig. SFV-host protein interactors identified through a genome-wide fluorescence microscopy based siRNA screen as exhibiting potential roles in SFV replication are additionally depicted by green and turquoise borders. A protein was defined as having a potential antiviral or proviral role if their Infection Index (II) values were ≥ 1.3 or ≤ 0.5, respectively. **b,** Tables showing the II values of SFV-host interactors with potential proviral or antiviral roles.

An 'Infection Index' (II) value was calculated for each gene depletion. This value indicated the fold change of infection upon depletion of the respective gene product, compared to the control, non-specific siRNAs (set as 1). A low multiplicity of infection (MOI) of 0.3 was used to allow for detection of both reduced or increased infection levels. The maximum II obtained in the screen was 1.85 (S2 Table). We therefore chose to set an II threshold of 1.3 to identify proteins having a potential antiviral role against SFV, and an II threshold of 0.5 to indicate proteins having a potential proviral role for SFV (S2 Table). When we compared the proteins identified in the siRNA screen with the SFV-host protein interaction networks, a sizable fraction of the interactors overlapped. Those with potential pro- or antiviral roles during infection are depicted with turquoise or green outlines, respectively (Figs 3A and S3) and collected in two lists: 'Proviral' and 'Antiviral', with their II values shown alongside them (Fig 3B). As expected, all the identified ribosomal proteins affecting SFV replication had a proviral effect (S3 Fig) and are listed separately (Fig 3B). In addition to ribosomal proteins, many other RNA binding proteins were also identified as having a potential role in SFV replication (Fig 3A and 3B [violet circles]). Those with the strongest proviral effects included the mRNA transport protein, nuclear RNA export factor 1 (NXF-1), as well as the splicing regulator, Splicing factor, proline- and glutamine-rich (SFPQ). Those with the strongest antiviral effects included proteins involved in mRNA turnover (Up-frameshift protein 1 [UPF1] and Exosome component 10 [EXOSC10]) and translation initiation (eukaryotic initiation factor subunit 1 [EIF2S1], aka eIF2α). Notably, more than 50% of these proteins have assigned functions in rRNA processing or ribosome biogenesis (Fig 3A and 3B [dark pink circles]).

We reasoned that analysing protein complexes that could exist or form between SFV protein interactors could give us further insight into how the virus could be influencing cellular function. We therefore performed protein-protein interaction enrichment analysis on the full list of SFV interactors (ribosomal proteins included) using the Metascape online tool (www. metascape.org), which incorporates various protein complex databases (see Material and Methods and Fig legend 4). From this, the integrated Molecular Complex Detection (MCODE) algorithm identified eight MCODE networks (Fig 4A). GO enrichment analysis of the MCODE networks revealed biological functions related to translation and NMD (MCODE1), ribosome biogenesis (MCODE2 and MCODE7), chaperones (MCODE3 and MCODE5), actin cytoskeleton organisation and mRNA splicing (MCODE4), and regulation of mRNA stability (MCODE6) (Fig 4A). This analysis corroborated what we observed through manual curation (Fig 3). Interestingly, each MCODE network (except MCODE8) included at least one SFV interactor that was identified as having a potential influence on SFV replication (Fig 4A). Since Metascape allows for the input of multigene lists, the protein-protein interaction enrichment analysis, followed by the MCODE algorithm was, in addition, applied independently to the lists of interactors for each individual SFV protein. GO enrichment analysis of the set of MCODE networks for each individual list (nsP1, nsP2, nsP3-Z, nsP4, capsid and Env) as well as for the merged list (All baits) was also applied. This allowed us to compare the significance of the GO terms collected in Fig 4A for the different SFV proteins (Fig 4B). We observed that the most significantly enriched GO terms were those related to translation, NMD and ribosome biogenesis. In addition, it became clear that it was mainly the interactors of nsP2, nsP3-Z and capsid that contributed to the enrichment of these GO terms (Fig 4B). Ribosome biogenesis related GO terms were most highly enriched for the capsid interactors (Fig 4B).

Because of the enriched GO terms, we chose to investigate the effect of SFV on translation, NMD and ribosome biogenesis. We reported previously that the NMD machinery could target the SFV genome independently of the 3´ UTR [33]. Half-life measurements of the genome of a replication incompetent SFV mutant suggested that this occurred early during infection, upon

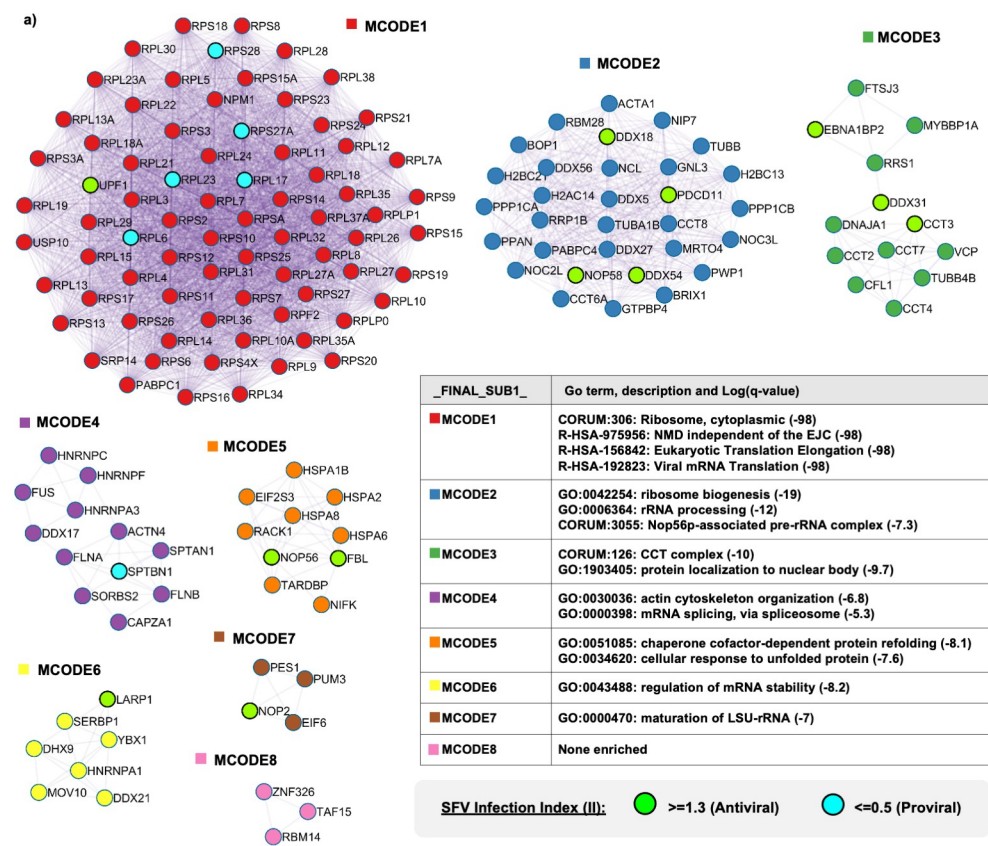

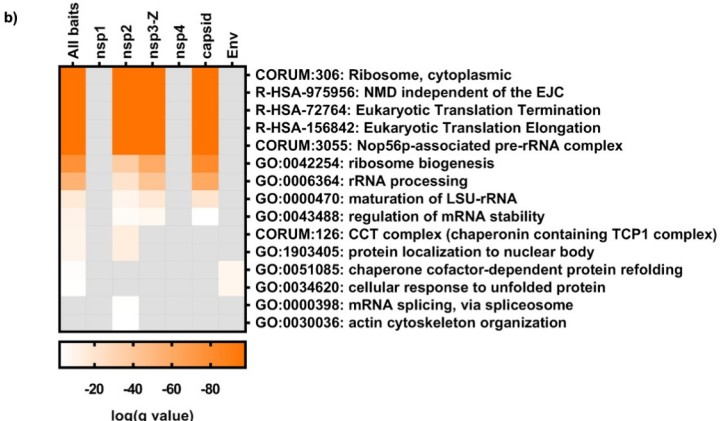

**Fig 4. SFV interactors are highly enriched for GO terms related to translation, NMD and ribosome biogenesis. a,** Protein-protein interaction enrichment analysis for the full list of SFV interactors was carried out using Metascape with the following databases: BioGrid, InWeb_IM and OmniPath. Densely connected network components were identified by applying the Molecular Complex Detection (MCODE) algorithm and the resultant MCODE networks are displayed. In addition, the proteins identified as having an effect on SFV replication through the siRNA screen are shown within the MCODE complexes (green and turquoise circles). In order to assign biological meanings to the MCODE networks, GO enrichment analyses were performed using the following sources: KEGG Pathway, GO Biological Processes, Reactome Gene Sets, Canonical Pathways and CORUM. All genes in the genome were used as the enrichment background. Between 1–4 GO terms were chosen to represent each MCODE network, with the relevant significance values depicted alongside them. **b,** Heat map comparing the significance of GO terms (chosen in a) among the different SFV baits. GO enrichment analysis was applied to the set of MCODE networks identified for each individual list (nsp1, nsp2, nsp3-Z, nsp4, capsid and Env), as well as for the merged list (All baits) and the significance values depicted in the heat map.

entry of the viral genome into cells [33]. Since viruses are known to evade cellular defence responses against viral infection, we wondered whether the virus could inhibit NMD at later stages of infection. Since NMD depends on translation [34] and viruses are known to inhibit translation, it was important to carefully analyse the time course of infection in our system in an attempt to disentangle these two tightly linked cellular processes. We used anti-ZsG or anti-nsP3 antibodies to detect nsP3-Z, as a representative for the presence of early produced non-structural proteins expressed from the gRNA, and anti-capsid antibodies to detect the capsid, as a representative for the presence of structural proteins that are expressed from sgRNAs later during the virus replication cycle (Fig 5A and 5B). The nsP3-Z protein was reproducibly detected at 3–4 hours post infection (p.i.) and as early as 2 hours p.i., while the capsid was reproducibly detected at 4 hours p.i. (Fig 5A and 5B). We measured the presence of phosphorylated (p-)eIF2α compared to total eIF2α as an indication of virus-induced translation inhibition and showed that a virus-dependent accumulation of p-eIF2α was reproducibly detected at 3–4 hours p.i. (Fig 5A). In addition, we performed time course puromycin incorporation assays to assess global translation activity using a more direct method [35]. This assay involves a puromycin pulse for 10 minutes, which causes the release of nascent polypeptides and results in many puromycin-labelled polypeptides of different lengths that can then be visualised by western blotting using anti-puromycin antibodies. The decrease in puromycin 3–4 hours p.i. is therefore indicative of a decrease in global translation, in agreement with the observed increase in p-eIF2α (Fig 5B).

We set aside samples from the time course infections in Fig 5A to assess NMD activity by measuring relevant RNA levels by RT-qPCR. To assess NMD activity, we adapted an assay described in [36], which measures the relative amounts of a NMD-sensitive splice isoform (NMD target) versus a NMD insensitive protein coding isoform (non-NMD target) of the same gene. We showed that at 4 hours p.i., the NMD-sensitive isoforms of both Hnrnpl (Hnrnpl_NMD) and BAG1 (BAG1_NMD) increased compared to the respective mRNA levels in uninfected cells, while the NMD insensitive isoforms (Hnrnpl_PROT and BAG1_PROT) remained relatively stable (Fig 5C). Notably, Hnrnpl_NMD already started to accumulate at 3 hours p.i., whereas the increase of BAG1_NMD only became apparent at 4 hours p.i. (S4 Fig). In addition, we measured the RNA levels of the well-known endogenous NMD targets, RP9P, IRE1α and GADD45, which also accumulated 3–4 hours p.i. (Figs 5C and S4). Together, these data are indicative of reduced NMD activity 3–4 hours p.i., suggesting that SFV can indeed inhibit NMD at later stages of infection. Since the timing of the NMD inhibition correlated with that of eIF2α-dependent inhibition of cellular mRNA translation, we were unable to pull apart the effect the viral infection had on the two cellular processes independently. Translation inhibition by SFV, and RNA viruses in general, is well described to occur through induction of p-eIF2α, which occurs upon host cell detection of the double-stranded viral RNA intermediate that arises during its replication cycle [10,14,15]. We therefore reasoned that, if one of the SFV proteins was responsible for the NMD inhibitory phenotype, we would be able to disentangle the effect of the virus on the two cellular processes. Taking the mass spectrometry data analysis (Fig 4B) into account, we reasoned that nsP2, nsP3-Z or capsid could be responsible for the NMD inhibitory phenotype. First, in order to confirm the interactions identified between nsP2, nsP3-Z and capsid with cellular UPF1 (Fig 3), we performed a reciprocal UPF1 immuno-precipitation (IP) from SFV infected cell lysates harvested 4 hours p.i. (Fig 5D). Mass Spectrometry allowed us to analyse the abundance of NMD factors and SFV proteins in the UPF1 eluates of uninfected (Un) and infected (4 hrs p.i.) cell lysates. The abundance of these proteins as a percentage of the UPF1 bait is depicted in the heat map (Fig 5D). UPF2 and UPF3B were ~30% as abundant as UPF1, indicating the success of the UPF1 IPs. UPF3A (in both) and SMG6 (in the uninfected sample only) were detected at lower abundances in the UPF1 IPs.

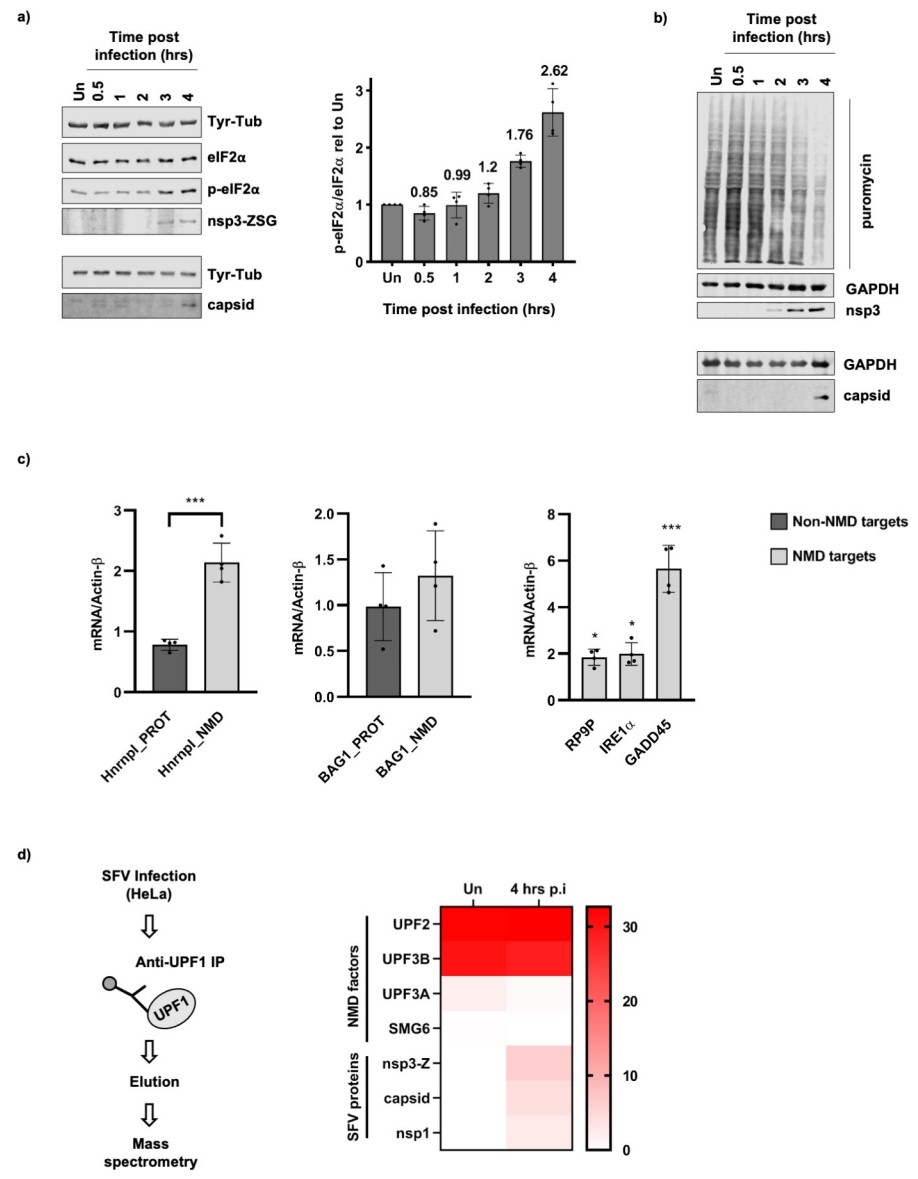

**Fig 5. SFV infection inhibits translation and NMD, and UPF1 associates with SFV proteins. a,** Representative western blots showing the induction of phosphorylated (p-)eIF2α versus eIF2α, as well as the accumulation of SFV proteins from the first ORF (nsp3-Z) and the second ORF (capsid) of the viral genome in a time course infection. SFV-ZSG infected HeLa cells were harvested at 0.5, 1, 2, 3 and 4 hours post infection. Uninfected cells (Un) were harvested at the 4 hour time point, Tyr-Tub was used as a loading control. The upper four panels represent a single blot that was cut and probed, while the lower two panels indicate a separate blot, on which the same set of samples was used. The bar graph (right) indicates the ratio of p-eIF2α/eIF2α relative to the uninfected sample, calculated from densitometry measurements of four western blots from independent time course infections. Mean values are shown above each bar (n = 4) and error bars reflect SD. **b,** Western blots showing the incorporation of puromycin into nascent polypeptides during a time course infection with SFV-ZSG (MOI = 10). GAPDH was used as a loading control. Nsp3-Z and capsid indicate the accumulation of SFV proteins during the infection. The upper 3 panels represent a single blot, while the lower two panels are from a separate blot analysing the same set of samples. **c,** RT-qPCR analysis showing mRNA levels relative to Actin-β of NMD targets (light grey bars) and non-NMD targets (dark grey bars) at 4 hours post infection, normalised to the uninfected control. Bar graphs display the mean ± SD (n = 4). Statistical significance was determined using unpaired, two-tailed t-tests, corrected for multiple comparisons using the Holm-Šídák method (GraphPad Prism v8.4.1), with alpha = 0.05. Asterisks indicate adjusted p values: p≤0.05*, p≤0.001***. **d,** Flow chart (left) depicting experimental approach for UPF1 immunoprecipitations (UPF1 IP) in uninfected (Un) and Infected (4 hours p.i.) cells, followed by mass spectrometry analysis to confirm UPF1 as an interactor of SFV proteins. Heat map (right) showing the abundance as % of the UPF1 bait for NMD factors and SFV proteins detected by mass spectrometry in the UPF1 eluate samples. Normalised spectral abundance factor (dNSAF) values were used.

The changes in levels of UPF3A and SMG6 detected in the uninfected sample compared to the infected sample were small and their significances remain unclear. The SFV proteins, nsP3-Z, capsid and nsP1 were identified in the UPF1 eluates of the infected sample (Fig 5D). It should be noted that their abundance was higher than that of the NMD factors, UPF3A and SMG6. Taken together, we were therefore able to confirm the interactions of nsP3-Z and capsid with UPF1.

The results above suggested that nsP3-Z and capsid were the most likely candidates to influence NMD activity in cells. Nevertheless, we decided to analyse the effect of all individual SFV proteins on NMD activity. To do this, the SFV proteins (from plasmids described in Fig 1B) were transiently expressed in HeLa cells (Fig 6A) and the relevant RNAs for each sample were measured by RT-qPCR analysis (Fig 6B). A striking and significant increase in the levels of NMD targets was observed upon expression of the capsid protein, while the non-NMD targets remained relatively stable (Fig 6B). No other significant changes in the levels of RNAs were observed upon expression of any of the other SFV proteins (Fig 6B). This data indicated that expression of the SFV capsid alone was sufficient to suppress NMD in cells. Since translation-related GO terms were also enriched for capsid interactors (among other SFV proteins) (Fig 4B), it was important to investigate whether expression of capsid influenced translation in cells, as this would in turn influence NMD activity. We showed that expression of capsid did not induce p-eIF2α (Fig 6C) nor, as judged from the puromycin incorporation assays, effect changes in global translation (Fig 6D). In addition, polysome profile gradients revealed that cells expressing the SFV capsid retained intact polysomes (S5 Fig), indicative of unperturbed translation. These three independent sets of data convincingly show that expression of the capsid protein did not influence global translation in cells. We therefore concluded that the SFV capsid suppresses NMD through a mechanism independent of translation inhibition. Since ribosome biogenesis related GO terms were most highly enriched among capsid interactors compared to the other SFV proteins, we used capsid expressing cells to look for any indication of altered ribosomes/rRNA that could give us any hints on the mechanistic action of capsid. However, we did not find any indications for capsid-induced defects in ribosome biogenesis or decreased ribosome abundance: 18S rRNA, 28S rRNA and 45S precursor rRNA as well as polysome gradients were not altered (S5 Fig).

## Discussion

A better understanding of the 'arms race' between viruses establishing a productive infection and the evolution of host cell protective mechanisms against them could aid in the development of antiviral strategies. Thorough investigations of virus-host interactomes provide crucial resources in advancing this knowledge. This study presents the first systematic virus-host protein interactome of the alphavirus, SFV. In addition to confirming previously identified host factors [18,29–31], our approach revealed many novel SFV-cell interactions. Many of the identified interactions were mediated by RNA. By comparing the results of the interactome with a siRNA screen against SFV host factors, we pinpointed pro- or antiviral activity to numerous newly identified cellular proteins. Below, we provide an overview on the interactors identified and discuss the potential role of these cellular proteins in the context of a viral infection.

### CCT complex

Subunits of the CCT complex were among the host interactions that stood out in the SFV-host protein interactome map (Fig 3). We identified seven members of the CCT complex, six of which (CCT2, CCT3, CCT4, CCT6A, CCT7 and TCP1) were previously found in RCs [18]. Our data revealed that the seven CCT complex members detected interacted with the viral helicase nsP2. Very little is known of the host partners of the viral RNA-dependant RNA

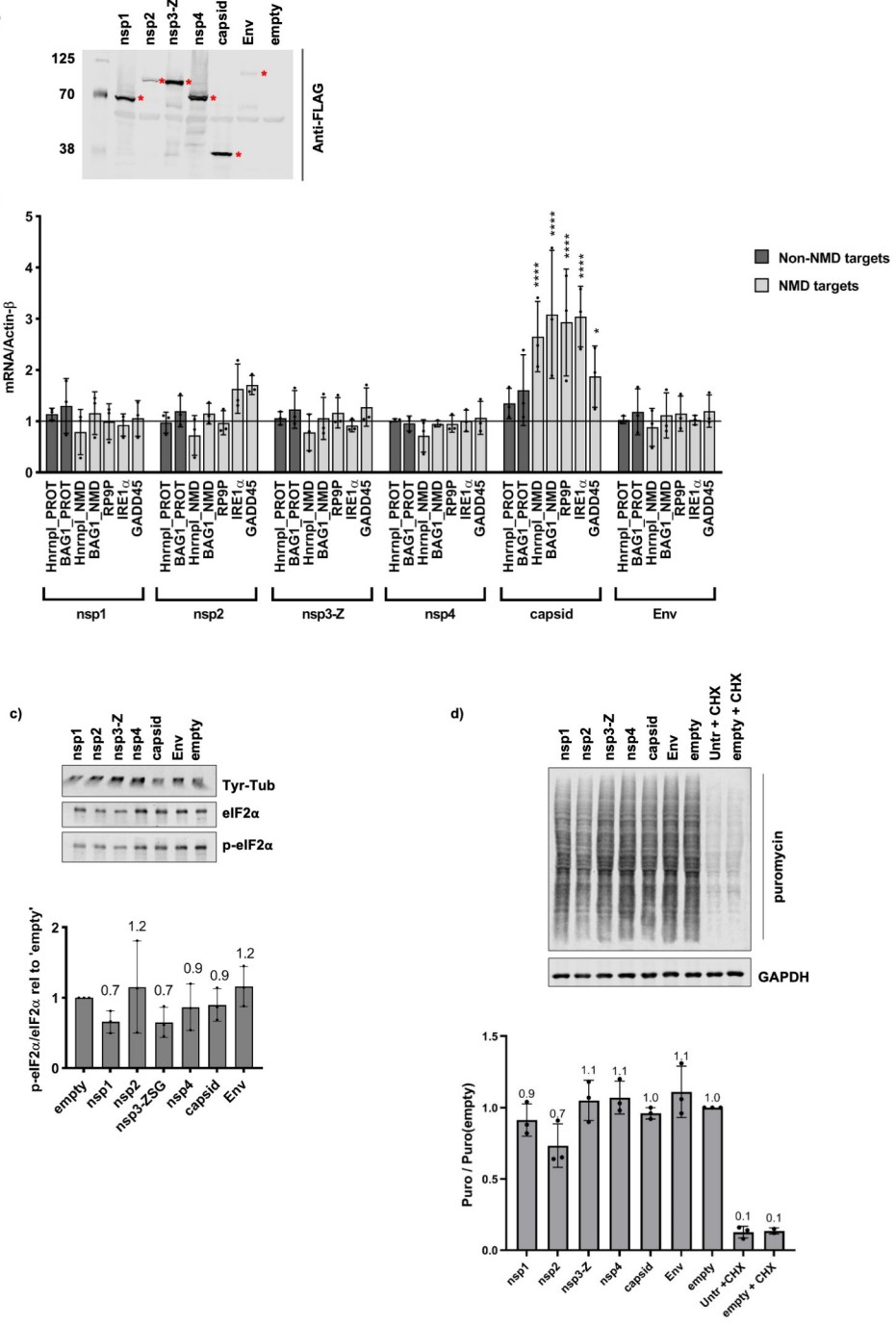

**Fig 6. SFV capsid protein inhibits NMD without affecting translation. a,** Anti-FLAG western blot showing expression levels of N-terminally 3xFLAG-tagged SFV proteins in HeLa cells. Red asterisks indicate the proteins at their expected sizes. **b,** RT-qPCR analysis showing mRNA levels relative to Actin-β of NMD targets (light grey bars) and non-NMD targets (dark grey bars) upon expression with the different SFV proteins, normalised to the 'empty' control. Bar graph displays the mean ± SD of three independent biological replicates. Statistical significance was determined using a 2-way ANOVA for multiple comparisons, followed by the Dunnett's multiple comparison test (alpha = 0.05) (GraphPad Prism v8.4.1). Asterisks indicate adjusted p values: $p \leq 0.05^{*}$, $p \leq 0.0001^{****}$. **c,** Western blot (top) showing p-eIF2α versus eIF2α upon expression of the SFV proteins. Tyr-Tub was used as a loading control. The bar graph (bottom) indicates the ratio of p-eIF2α/eIF2α relative to the 'empty' control. The ratios were calculated from densitometry measurements of three western blots of independent transfection experiments. Mean values are shown above each bar (n = 3) and error bars indicate SD. **d,** Western blot showing puromycin incorporation, indicative of ongoing translation, in SFV-protein expressing cells. Cycloheximide (CHX) was used as a positive control for global

translation inhibition, GAPDH was used as a loading control. The bar graph (bottom) indicates the quantification of the puromycin signal relative to the puromycin signal of the 'empty' control transfection. Mean values are shown above each bar (n = 3) and error bars indicate SD.

polymerase nsP4. We identified CCT3 and CCT8 as specific nsP4 interactors. These results suggest that nsP2 and nsP4 could be responsible for recruiting the CCT complex to RCs during infection. The CCT complex, known to assist in the correct folding of cellular proteins such as actin and tubulin [37–39], could play a crucial role in guiding the folding of the two long polyprotein precursors that, after proteolytic cleavages, generate all the individual viral proteins. The CCT complex could contribute to the shift of the nsPs in their relative stoichiometric compositions during the course of the infection by differentially stabilising individual nsPs. CCT3 was additionally identified in the siRNA screen. The CCT complex could therefore present an interesting novel antiviral target.

## Cytoskeleton

Interestingly, nsP2 interacted with a large number of cytoskeleton or cytoskeleton signalling proteins, of which ACTN4 was highly abundant in the nsP2 pulldown sample and enriched upon treatment with RNase A, indicating ACTN4 may bind directly to nsP2. Additionally, tubulins (TUBB, TUBA1B, TUBB4B) were the only non-ribosomal proteins identified as nsP1 interactors. NsP1 has previously been reported to anchor the RC to the PM and to induce formation of filopodia-like structures, concomitant with a rearrangement of cytoskeletal proteins [20]. Since during the course of an infection, nsPs form RCs within dynamic 'spherule' structures [9,11], the binding of nsPs to cytoskeletal proteins and their chaperones may play a crucial role in reorganising the host cytoskeletal network, thereby enabling the RCs to migrate. Cytoskeleton/cytoskeletal signalling interactors additionally identified in the siRNA screen include ANXA2, CSRP2, FLNC and SPTBN1. Further studies are needed to validate these interactions and elucidate mechanisms by which SFV proteins could exploit the host cytoskeleton or hijack the CCT complex to aid in the infection.

## ER chaperones

The specificity of our approach was also confirmed by the unique interactors of the viral envelope protein (Env), a transmembrane trimer that folds and is post-translationally modified in the lumen of the secretory pathway [17]. These include a number of ER chaperones (BAG2, BAG6, HSPA1B, HSPA2, HSPA6, HSPA8) and the transitional ER ATPase, VCP.

## RNA binding proteins

The nsP2, nsP3-Z and capsid proteins were found to mediate many of their host interactors through RNA (Fig 2A and 2C). As such, a large number of RNA-binding proteins were identified as host interactors of these three SFV proteins (Fig 3A), raising the question of whether they could play a role in altering and exploiting the compositions of mRNPs during infection. Some of the identified RNA-binding proteins that were previously found in SFV RCs include HNRNPC, HNRNPA1, SFPQ, DHX9, DDX3X, PABPC1, G3BP1 and G3BP2 [18]. In addition to being found in RCs, the nsP3:G3BP interaction has been well characterised [29–31]. SFV nsP3 has been reported to bind G3BP and suppress the formation of stress granules, thought to have antiviral activity [30]. Consistent with these previous findings, we identified G3BP1, G3BP2 and USP10, a deubiquitinase protein known to bind G3BP [30], as interactors of nsP3-Z. USP10 was identified as a unique interactor of nsP3, while G3BP1 and G3BP2 were

identified as also interacting with nsP2 and capsid. Thanks to a nuclear translocation signal, nsP2 shuttles between cytoplasm and nucleus during infection and interferes with transcription [22]. In line with this, and as observed by others, we observed localisation of the 3xFLAG-nsP2 in the nucleus of HeLa cells at steady state. Our study identified a number of nuclear proteins interacting with nsP2, many of which are splicing regulators, including HNRNPC, HNRNPA1, HNRNPA3, HNRNPF, HNRNPH3 and SFPQ. HNRNPC was the most abundant significantly enriched protein in the nsP2 pulldown. It was also significantly enriched and abundant in the RNaseA-treated nsP2 sample, indicating that this interaction may be either direct or mediated by another protein. Interestingly, SFPQ was found to be one of the most abundant interactors not only of nsP2, but also of capsid (Fig 2C). SFPQ was additionally identified through the siRNA screen, along with the mRNA export factor, NXF-1, as being among the strongest proviral interactors (i.e. depletion of these factors inhibited viral infection). The binding of nsP2 and capsid to these nuclear proteins could influence the regulation of RNA processing or mRNA modification steps in the nucleus or re-localise nuclear proteins to the cytoplasm in order to achieve a productive infection. Some cytoplasmic viruses, for example, hijack nuclear proteins including splicing factors (hnRNPs and SFPQ) from the nucleus to the cytoplasm, increasing infectivity [40–42]. We also identified RNA-binding interactors exhibiting strong antiviral effects (i.e. depletion of these proteins enhanced viral infection). These included UPF1, EXOSC10 and EIF2S1 (also known as eIF2α). We have previously validated the antiviral role of UPF1, the master regulator of NMD [33]. Here we have identified candidate viral proteins that might counteract cellular intrinsic antiviral functions.

### Ribosomal and ribosome biogenesis factors

Notably, 77 of the 251 identified host cell interactors were ribosomal proteins. Consistent with reports indicating the association of SFV capsid [43–45] and other alphavirus nsP2 and nsP3 [31,46] with ribosomal subunits, ribosomal interactors identified came mainly from nsP2, nsP3-Z and capsid affinity purifications (Fig 2D). Some of these exhibited proviral activity, including RPS27a, which was interestingly the only ribosomal protein to be found bound to (in addition to nsP2) nsP1, nsP4 and Env (Figs 2C and S3). We were surprised by the presence of newly identified nuclear interactors involved in rRNA processing and ribosome biogenesis, many of which exhibited antiviral activity. Interestingly, many of these were uniquely bound to capsid (Fig 3A). Evidence of capsid in the nucleus has previously been reported [47] and we were able to trap 3xFLAG-capsid in the nucleus of HeLa cells upon blocking of export. Even so, little is known about the role of the capsid in the nucleus and how this could affect the cellular ribosome. We therefore assessed polysome gradients (S5 Fig) and measured 18S, 28S and 45S precursor rRNAs in nuclear fractions of capsid expressing cells, but found no obvious phenotypic changes compared to cells expressing the 'empty' vector. Perhaps the effects of these interactions on the ribosome are more subtle or only affect a small pool of 'specialised' ribosomes, making changes difficult to detect. Since viruses rely on the host cell ribosome for translation of their own genomes, a better understanding of the involvement of the viral proteins in recruiting or potentially altering the host cell ribosomes through interaction with specific ribosomal proteins and this novel set of ribosome biogenesis factors definitely warrants further investigation.

To obtain additional hints about possible functional consequences of the detected interactions between SFV and host cell proteins, we used MCODE to analyse the protein complexes that could form between all host interactors, including the ribosomal proteins (Figs 4 and S3). GO enrichment analyses of the protein complexes reinforced many of the cellular processes discussed above and revealed that the most highly enriched GO terms were related to translation and NMD (Fig 4A and 4B). As a counter defence strategy, viruses are known to inhibit

cellular mRNA decay factors that can degrade viral RNAs and restrict infection [32,48,49]. We therefore hypothesized and decided to investigate, whether SFV was able to inhibit the NMD pathway, which has antiviral activity against alphaviruses [33]. Indeed, we found that starting from 3–4 hours after infection, SFV antagonises the NMD pathway, with consequent stabilisation of *bona fide* NMD mRNA transcripts (Figs 5C and S4). Viruses known to inhibit mRNA decay pathways do so by different mechanisms. Often a viral protein counteracts a key cellular regulator. Here, we show novel data that in the case of SFV, it is the capsid protein that inhibits NMD (Fig 6B). Therefore, inhibiting this function of the viral capsid could lead to novel avenues for therapeutic intervention.

Using both SFV protein affinity purifications in transient expression experiments and UPF1 IPs in SFV infected cells, we show that the core NMD factor UPF1 binds to SFV capsid among other SFV proteins in an RNA-dependent manner. Together, this indicates that the capsid, and potentially other SFV proteins, associate with mRNP molecules that also contain UPF1. The large number of ribosomal proteins pulled down by capsid (Figs 2D, 4B and S3) could indicate that UPF1 and capsid are perhaps sitting on ribosome-associated mRNPs. Further studies are required to elucidate the mechanism of NMD inhibition. Thus far, evidence for NMD suppression by the capsid proteins of the *Coronavirus*, Mouse Hepatitis Virus (MHV) as well as the *Flavivirus*, Zika Virus (ZIKV) have been reported [50,51]. It was also postulated that the capsid or 'core' protein of Hepatitis C Virus (HCV) may be responsible for the NMD inhibitory phenotype that was reported upon HCV infection [52]. Our findings therefore add SFV as the first alphavirus to a growing list of viruses of which the capsid protein is responsible for an NMD inhibitory effect. Though the stability of the SFV genome has thus far been attributed to evasion of deadenylation through binding to HuR [53], the virus may require additional strategies to protect itself in order to ensure efficient translation of viral genes and packaging of genomes into new progeny viruses. Perhaps the SFV capsid plays a protective role against degradation of its RNAs by NMD.

In summary, we present here two valuable resources that will aid in the study of SFV: a SFV-host protein interactome as well as a genome-wide siRNA screen for host factors influencing SFV infection. Interestingly, some of the SFV host interactors we identified have been identified for other alphaviruses through methods including the recombinant tagging of one of the viral proteins during infection followed by imaging or affinity purifications and Yeast-2-Hybrid assays [46,54–57]. For example, G3BP1, G3BP2, FXR1, FXR2, CAPZB and CAPZA1 have also been found to interact with nsP3 of Venezuela Equine Encephalitic virus (VEEV) [55] and HNRNPC and HNRNPA3 have been found to interact with nsP2 of CHIKV [46]. Noteworthy, the Yeast-2-Hybrid screen used to elucidate CHIKV-host protein interactions would not have detected RNA-mediated interactions [46]. We therefore believe our study provides a valuable resource not only for the study of SFV, but also for the further study of other related alphaviruses. Our results suggest that RNA binding proteins, so far not investigated in the context of virus infections, play crucial roles in SFV infectivity. SFV-host protein-protein interactions for many of these proteins occurred through an RNA substrate, indicating that the SFV proteins are likely binding to the same mRNPs. Analysing changes in compositions of cellular mRNPs during the course of infection may provide mechanistic insight into how SFV proteins may be influencing the landscape of host mRNPs, in turn affecting or exploiting RNA processes such as transport, splicing, degradation and translation.

## Material and methods

### Plasmids

The coding sequences for the SFV proteins were cloned into pcDNA5_FRT_TO_3xFlag(N), to yield pcDNA5.3xFLAG-nsp1, pcDNA5.3xFLAG-nsp2, pcDNA5.3xFLAG-nsp3-ZSG,

pcDNA5.3xFLAG-nsp4, pcDNA5.3xFLAG-capsid and pcDNA5.3xFLAG-Env, which were used for all subsequent transfections. pcDNA5_FRT_TO_3xFlag(N) was linearised using *Bam*HI. PCR products for each of the SFV proteins, were generated from either SFV-ZSG (-3'UTR) [33], SFV-capsid or SFV-Envelope [58] plasmids using the following primers:

| nsp1: | 5'-CGACAAGCTTGGATCCGCCGCCAAAGTGCATGTTGAT-3' and |
| | 5'-CTGGACTAGTGGATCTTATGCACCTGCGTGATACTCTAGTT-3' |
| nsp2: | 5'-CGACAAGCTTGGATCCGGGGTCGTGGAAACACCTC-3' and |
| | 5'-CTGGACTAGTGGATCTTAACACCCGGCCGTGTGCA-3' |
| nsp3-ZSG: | 5'-CGACAAGCTTGGATCCGCACCATCCTACAGAGTTAAGAGAGCAG-3' and |
| | 5'-CTGGACTAGTGGATCTTATGCACCCGCGCGGCCTA-3' |
| nsp4: | 5'-CGACAAGCTTGGATCCTATATTTTCTCCTCGGACACTGGCA-3' and |
| | 5'-CTGGACTAGTGGATCTTAACGCACCAATCTAGGACCG-3' |
| capsid: | 5'-CGACAAGCTTGGATCCAATTACATCCCTACGCAAACGTTT-3' and |
| | 5'-CTGGACTAGTGGATCCTACCACTCTTCGGACCCCT-3' |
| Env: | 5'-CGACAAGCTTGGATCCTCCGCCCCGCTGATTACTG-3' and |
| | 5'-CTGGACTAGTGGATCTTATCTGCGGAGCCCAATGCAAG-3' |

Primers were designed such that each PCR product contained 15-bp homology arms complementary to the ends of *Bam*HI-linearised pcDNA5_FRT_TO_3xFlag(N). The respective PCR products were then cloned into pcDNA5_FRT_TO_3xFlag(N) using In-Fusion HD Cloning Kit (Cat. No. 639650, Takara).

## Antibodies and dilutions

Primary antibodies: Mouse monoclonal anti-FLAG M2 (1:2500) (Sigma Aldrich, Cat#: F3165); Mouse monoclonal anti-Tyrosine-Tubulin (1:10000) (Sigma Aldrich, Cat#: 9028); Rabbit polyclonal anti-eIF2α (1:1000) (cell signalling technology, Cat#: 9722); Mouse monoclonal anti-eIF2α, L57A5 (1:1000) (cell signaling technology, Cat#: 2103); Rabbit polyclonal anti-p-eIF2α (Ser51) (1:1000) (cell signaling technology, Cat#: 9721); Rabbit anti-nsp3 (1:3000) [9]; Mouse monoclonal anti-ZsGreen1 (ZSG) clone TI2C2 (1:2000) (OriGene, Cat#: TA180002); Mouse anti-capsid (1:3000) [33]; Mouse monoclonal anti-puromycin, 12D10, (1:15000) (Millipore, Cat#: MABE343); Rabbit polyclonal anti-GAPDH (FL-335) (1:1000) (Santa Cruz Biotechnology, Cat#: sc-25778); Goat polyclonal anti-RENT1 (UPF1) (1:1000) (Bethyl, Cat#: A300-038A). Secondary antibodies (1:10000) (LICOR): Donkey anti-goat 800CW; Donkey anti-rabbit 800CW; Donkey anti-mouse 800CW; Donkey anti-rabbit 680LT; Donkey anti-mouse 680LT.

## Cell culture

HeLa cells were cultured in Dulbecco's Modified Eagle Medium (DMEM) supplemented with FCS, Penicillin and Streptomycin (DMEM +/+) at 37˚C under 5% carbon dioxide atmosphere. Passaging and harvesting of cells were done by detachment using Trypsin/EDTA solution at approximately 1:10 (v/v) of the culture volume. Cells were quantified using trypan blue staining followed by automated cell counting using the Countess Automated Cell Counter (Thermo Fisher Scientific).

## SFV protein expression and affinity purification

HeLa cells were seeded ($2.5 \times 10^6$ per dish) into 15 cm dishes. The following day, 20.8 μg of the relevant pcDNA5.3xFLAG plasmids (described above) were transfected using Dogtor

transfection reagent (OZ Biosciences), as per manufacturer's recommendations. Cells were harvested 48 hours later, collected by centrifugation (250 x g, 4˚C, 5 minutes), washed once with 1xPBS, and stored at -80˚C until use. 500 μL of lysis buffer (150 mM NaCl, 0.5% Triton X-100, 50 mM HEPES [pH 7.4], 1x protease inhibitor cocktail [Biotool]) was added to each cell pellet and resuspended by vortexing. Cells were lysed by sonication (2x10 second pulses, Amp = 45) and cell lysates cleared by centrifugation (16000 x *g*, 4˚C for 10 minutes). 12 μg anti-FLAG M2 antibody was coupled to 1.5 mg of magnetic Dynabeads M270 Epoxy (Thermo Fisher Scientific), as per manufacturer's instructions. The beads were washed with lysis buffer and 500 μL of the cleared cell lysate was added, and the mixture was then incubated at 4˚C for one hour with rotation. Beads were collected on a magnet and washed three times with lysis buffer. At the third wash, each sample was split into two (for treatment or no treatment with RNase A). For +RNase A samples, RNase A treatment was performed on the beads. The supernatant was removed using the magnet, and 50 μL of RNase A (0.8 mg/mL, Sigma Aldrich) containing lysis buffer was added to the beads and incubated at 25˚C for 15 minutes, shaking. Thereafter, the RNase A treated beads were washed with lysis buffer, and then samples eluted from the beads using 3xFLAG peptide (sciencepeptide.com): 20 μg of FLAG peptide was incubated with the beads at 25˚C for 15 minutes, shaking. The FLAG peptide elution was also performed for the–RNase A samples. Thereafter, the eluates were collected and 10 μL of loading buffer (4xLDS + DTT [75 mM]) was added to each eluate, while 30 μL of loading buffer was added to the remaining beads samples. The samples were then incubated at 75˚C for 10 minutes, ready for analysis by western blot, silver stain, and coomassie gel for mass spectrometry sample preparation.

## Silver stain

Samples were electrophoresed on 26-well NuPAGE 4–12% Bis-Tris gradient gels (Thermo Fisher Scientific) in 1xMOPS running buffer at 200V for approx. 1 hour. The gels were fixed in 50% methanol/12% acetic acid for one hour at room temperature, followed by three 5 minutes washes in 35% ethanol. The gels were sensitised for 2 minutes in 0.02% sodium thiosulfate, followed by three 5 minutes washes in Milli-Q H$_2$O. The gels were stained in 0.2% silver nitrate/ 0.03% formaldehyde for 20 minutes, followed by two 1 minute washes in Milli-Q H$_2$O. The gels were developed in 0.57 M sodium carbonate + 0.02% formaldehyde/0.0004% sodium thiosulfate and then incubated in 50% methanol/12% acetic acid for 5 minutes. Thereafter, the gels were placed in 1% acetic acid for short term storage at 4˚C. Images were taken using the gel documentation system (www.vilber.com)

## Coomassie gels and mass spectrometry analysis

The protein compositions of the eluates from the SFV affinity purifications were analysed by label-free quantitative mass spectrometry. The eluates (20 μL) were electrophoresed in 1xMOPS running buffer about 1 cm into the 26-well NuPAGE gels and then stained with coomassie-blue (10% phosphoric acid, 10% ammonium sulfate, 0.12% coomassie G-250, 20% methanol) as described previously [59]. Images were taken using the gel documentation system. Rectangular segments (10 mm x 3 mm) for each lane were cut from the gels using sterile blades. The gel pieces were reduced, alkylated and digested by trypsin as described elsewhere [60]. The digests were analysed by liquid chromatography (LC)-MS/MS (PROXEON coupled to a QExactive HF mass spectrometer, ThermoFisher Scientific), injecting 5 μL of the digests. Peptides were trapped on a μPrecolumn C18 PepMap100 (5μm, 100 Å, 300 μm×5mm, ThermoFisher Scientific, Reinach, Switzerland) and separated by backflush on a C18 column (5 μm, 100 Å, 75 μm×15 cm, C18) by applying a 60-minutes gradient of 5% acetonitrile to 40%

in water, 0.1% formic acid, at a flow rate of 350 nL/min. The Full Scan method was set with resolution at 60,000 with an automatic gain control (AGC) target of 1E06 and maximum ion injection time of 50 ms. The data-dependent method for precursor ion fragmentation was applied with the following settings: resolution 15,000, AGC of 1E05, maximum ion time of 110 milliseconds, mass window 1.6 m/z, collision energy 27, under fill ratio 1%, charge exclusion of unassigned and 1+ ions, and peptide match preferred, respectively. Proteins in the different samples were identified and quantified with MaxQuant [61] against the human swissprot [62] database (release June 2019), in addition to custom nsP1, nsP2, nsP3-Z, nsP4, capsid and Env sequences. The following MaxQuant settings were used: separate normalisation groups for the +Rnase A and -Rnase A samples, mass deviation for precursor ions of 10 ppm for the first search, maximum peptide mass of 6000Da, match between runs activated with a matching time window of 0.7 min only allowed across replicates; cleavage rule was set to strict trypsin, allowing for 3 missed cleavages; allowed modifications were fixed carbamidomethylation of cysteines, variable oxidation of methionines, deamination of asparagines and glutamines and acetylation of protein N-termini. Protein intensities are reported as MaxQuant's iTop3 [63] values (sum of the intensities of the three most intense peptides). Peptide intensities were first normalised by variance stabilisation normalisation and imputed. Imputation values were drawn from a Gaussian distribution of width 0.3 centred at the sample distribution mean minus 1.8x the sample standard deviation, provided there were at least 2 evidences in the replicate group. In order to perform statistical tests, iTop3 values were further imputed at the protein level, following the rule: 'if at least two detections in at least one group' and using the following protein impute parameters: imputation values were drawn from a Gaussian distribution of width 0.3 centred at the sample distribution mean minus 2.5x the sample standard deviation. Potential contaminants and proteins marked 'only identified by site' were removed prior to performing Differential Expression (DE) tests, which were done by applying the empirical Bayes test [64] followed by the FDR-controlled Benjamini and Hochberg [65] correction for multiple testing. A significance curve was calculated based on a minimal log2 fold change of 1 and a maximum adjusted $p$-value of 0.05. Proteins that were consistently reported as DE throughout 20 imputation cycles were flagged as "persistent". In addition, SAINT analysis was performed according to [66] and a significance threshold of FDR<0.05 was applied.

## Collection of SFV protein interactors through further filtering by abundance

Significant interactors for each SFV bait protein were defined as those that were significantly differentially expressed (enriched) compared to the untransfected control and the significance persisted throughout the imputation cycles. In addition, the proteins taken as "significant interactors" had to be considered "true interactors" as determined using SAINT analysis (with a threshold FDR of $\leq$ 0.05). To simplify the lists and attempting to retain potentially biologically relevant interactors, we further filtered the lists, retaining proteins whose abundance made up at least 0.5% of the relevant bait protein. In the case of the nsp3-Z bait, which was very lowly abundant in the sample as it proved difficult to elute from the beads, we retained proteins whose abundance made up at least 5% of the SFV bait.

## Interaction networks

The final lists of proteins were used to create SFV-host protein interaction networks using Cytoscape_v3.7.2. STRING analysis (string-db.org, version 11.0) was performed using a minimum required interaction score of highest confidence (0.900) for both database and

experimental evidence and the known protein-protein interactions were overlaid onto the SFV-protein interactome networks, using Cytoscape_v3.7.2.

### siRNA screen

The automated, image-based, genome-wide siRNA screen against cellular host factors involved in the infection of SFV was described previously [33]. Briefly, HeLa cells were transfected with siRNA at a final concentration of 20 nM in 384-well plates using 0.1 µL RNAiMAX (Life Technologies) per well in 100 µL cell culture medium. The siRNA library consisted of four pooled siRNA oligonucleotides per gene (Human ON-TARGETplus, Dharmacon). 72 hours after transfection, cells were infected for 6 hours with SFV-ZsG at a concentration giving an infection rate of approx. 30% in control siRNA-treated cells. Following fixation in 4% formaldehyde and Hoechst staining, nine images per well were acquired using high-content automated fluorescence microscopes (ImageX-press, Molecular Devices). Infected cells were detected using Cell Profiler (www.cellprofiler.org) and Advanced Cell Classifier (www.acc.ethz.ch/acc.html) and the % of infected cells per well was determined. An "Infection Index" value was calculated for each gene, indicating the fold change of infection upon depletion of the gene product, compared to the control siRNAs, which was set as 1. An Infection Index threshold of 1.3 was chosen to indicate proteins having a potential antiviral role against SFV, and an Infection Index threshold of 0.5 to indicate proteins having a potential proviral role for SFV (S2 Table, raw data). The proteins identified in the siRNA screen were overlaid with the SFV-host protein interaction networks.

### Protein-protein Interaction enrichment analysis/MCODE analysis

Protein-protein interaction enrichment analysis was performed using the Metascape online tool (www.metascape.org) according to [67]. Metascape allows for the input of multigene lists. For each given list (ie: interactors of nsp1, nsp2, nsp3-Z, nsp4, capsid and Env) as well as for the merged list, protein-protein interaction enrichment analysis was carried out using the following databases: Bio-Grid, InWeb_IM and OmniPath. The resultant networks contained the subset of proteins that form physical interactions with at least one other member in the list (PPI networks). If the network contained between 3 and 500 proteins, the Molecular Complex Detection (MCODE) algorithm was applied to identify densely connected network components, which were extracted from the PPI networks and displayed as individual 'MCODE' networks. These MCODE networks were identified separately for each individual list as well as for the merged list (Fig 4A). In order to assign biological meanings to the MCODE networks, Gene ontology (GO)/Pathway and Process enrichment analysis was applied to each MCODE network (FINAL_SUB1_MCODE1, 2,3,4,5,6,7) identified for the merged list of interactors. GO/pathway and process enrichment analysis was carried out with the following ontology sources: KEGG Pathway, GO Biological Processes, Reactome Gene Sets, Canonical Pathways and CORUM, using the Metascape tool. All genes in the genome were used as the enrichment background. From the top10 most significant GO term descriptions gathered for each MCODE network (S3 Table), between 1–4 terms were chosen to assign biological meaning to each MCODE term (Fig 4A). The log(q-value) of the terms indicates the significance calculated using the merged list of interactors (S3 Table and Fig 4A). This list of GO term descriptions was then used to compare these biological functions among individual SFV proteins (Fig 4B). To do this, GO/pathway and process enrichment analysis was additionally applied to the set of MCODE networks (MCODE_ALL) for the individual lists (nsp1, nsp2, nsp3-Z, nsp4, capsid and Env), as well as for the merged list (All baits) (FINAL_MCODE_ALL). From this analysis, the significance [log(q value)] values for the chosen list of GO term descriptions were then used to create a heat map (GraphPad Prism v8.4.1) (Fig 4B).

## SFV time course infection

HeLa cells were seeded ≤ 12 hours prior to SFV time course infection. Medium was aspirated and replaced with ice-cold infection medium (RPMI, 0.02 M HEPES [pH7.1], 0.2% BSA) and cells placed at 4˚C for 15 minutes in preparation for the infection. Medium was aspirated and cells were infected with SFV-ZSG in infection medium at an MOI of 10. In order to synchronise the infection, cells were placed at 4˚C for 45 minutes, with swirling every 5 minutes. The SFV-ZSG-containing media was aspirated, cells washed once with 1x PBS, warm infection medium was added and cells were incubated at 37˚C. Cells were harvested at 0.5, 1, 2, 3 and 4 hours post infection (post incubation at 37˚C). The uninfected control was also harvested after 4 hours incubation at 37˚C. During harvesting, cells for each condition were split in half, such that one half was collected for western blot analysis, and the other for RNA analysis. After the cells were collected by centrifugation at 4˚C for 5 minutes at 250 x $g$ and washed with 1x PBS, the cell pellets for western blot analysis were resuspended in 2x SDS-PAGE loading buffer and heated at 95˚C for 5 minutes, while those for RNA analysis were resuspended in 900 μL TRI-reagent for RNA extraction.

## Puromycin incorporation assay (time course infection)

HeLa cells were infected with SFV-ZSG as described above, with the following adjustment: 10 minutes prior to the relevant harvesting time point (0.5, 1, 2, 3 and 4 hours post infection), cells were pulse labelled for 10 minutes with 10 μg/mL puromycin followed by a 10 minutes recovery step with infection medium at 37˚C. Cells were harvested and washed pellets resuspended in 2x SDS-PAGE loading buffer and incubated at 95˚C for 5 minutes. Lysates were electrophoresed in 10% SDS-PAGE gels and analysed by western blot using mouse anti-puromycin antibodies.

## RNA extraction and RT-qPCR

Total RNA was extracted from TRI-reagent by isopropanol precipitation and resuspended in disodium citrate buffer (pH 6.5). Contaminating DNA was degraded by treatment with Turbo DNase (Ambion). Thereafter, reverse transcription was carried out using AffinityScript Multiple Temperature Reverse Transcriptase (Agilent), followed by qPCR using Brilliant III Ultra-Fast SYBR Green qPCR Master Mix (Agilent), according to manufacturer's instructions. The following primers were used:

| | |
|---|---|
| Hnrnpl_PROT | 5'- CAATCTCAGTGGACAAGGTG—3' |
| | 5'- CTCCATATTCTGCGGGGTGA—3' |
| Hnrnpl_NMD | 5'- GGTCGCAGTGTATGTTTGATG—3' |
| | 5'- GGCGTTTGTTGGGGGTTGCT—3' |
| BAG1_PROT | 5'- ACTCATATTTAAGGGAAAATCTCTG—3' |
| | 5'- TTGGGCAGAAAACCCTGCTG—3' |
| BAG1_NMD | 5'- CATATTTAAGGTTCTTCAACAGATA—3' |
| | 5'- TGTTTCCATTTCCTTCAGAGA—3' |
| RP9P | 5'- CAAGCGCCTGGAGTCCTTAA—3' |
| | 5'- AGGAGGTTTTTCATAACTCGTGATCT—3' |
| IRE1α | 5'- TGCAGGTCCCAACACATGTGG-3' |
| | 5'- TCAGGCCTTCATTATTCTTGC-3' |
| GADD45β | 5'- TCAACATCGTGCGGGTGTCG—3' |
| | 5'- CCCGGCTTTCTTCGCAGTAG—3' |
| Actinβ | 5'- TCCATCATGAAGTGTGACGT -3' |
| | 5'- TACTCCTGCTTGCTGATCCAC -3' |

The reaction and fluorescence readout was carried out in the Rotor-Gene 6200 (Corbett Life Science) real-time system. Threshold cycle values (ct-values) were set manually and the relative mRNA levels calculated using the comparative CT method. Graphs are displayed as the mean ± the standard deviation (SD), with n values indicated in the respective figure legends. For the time course infections, statistical significance was determined using unpaired, two-tailed t-tests, corrected for multiple comparisons using the Holm-Šídák method, with alpha = 0.05. For the transient transfections, statistical significance was determined using a 2way ANOVA for multiple comparisons, followed by the Dunnett's multiple comparison test (alpha = 0.05). Within each column [each qPCR assay], each SFV protein was compared against the "empty" control row. GraphPad Prism v8.4.1 was used for statistical analyses and to create the plots.

## UPF1 immunoprecipitation and mass spectrometry

HeLa cells were infected as described under "SFV time course infection" and harvested at 4 hours post infection. The uninfected control was also harvested at 4 hours post infection. After the cells were harvested by centrifugation at 4˚C for 5 minutes at 250 x *g* and washed with 1x PBS, the cell pellets were frozen and stored at –80˚C until use. Cell pellets were thawed on ice and a volume of 500 μL of lysis buffer (0.5% Np40 (IGEPAL), 150 mM NaCl, 50 mM Tris buffer [pH 7.5], 1xprotease Inhibitor cocktail [Biotool], RNase Inhibitor [6 μL RNase Inhibitor/mL of lysis buffer]) was added to each. The cell pellets were resuspended by vortexing and cells lysed by sonication: 3x 5 seconds pulses, 45% Amplitude, with cooling on ice between each sonication. Cell lysates were cleared by centrifugation at 4˚C for 10 minutes at 16000 x *g*. Goat anti-UPF1 antibody (Bethyl) was coupled to magnetic Dynabeads Protein G (Thermo Fisher Scientific). An amount of 1.5 mg of beads pellet per sample was resuspended in 200 μL PBS-T (0.02% Tween20) containing 12 μg antibody. The mixture was rotated for one hour at 4˚C. The coupled beads were washed three times with lysis buffer and then 500 μL of the cleared cell lysate was added to the beads pellet. The cleared lysates and coupled beads were incubated at 4˚C for one hour with rotation. The beads were collected on a magnet and washed three times with lysis buffer. At the third wash, each sample was split into two (for treatment or no treatment with RNaseA). The supernatant was removed and the beads were resuspended with 20 μL lysis buffer + 10 μL loading buffer (4x LDS +DTT). The samples were incubated at 75˚C for 10 minutes. Using the magnet, the supernatants (eluates) were transferred to new tubes, ready to load onto gels for coomassie staining and mass spectrometry sample preparation. The compositions of the eluates were quantified by mass spectrometry, following the same protocol as above. Triplicate samples were processed against the same sequence database as above, by Transproteomics pipeline (TPP) [68] tools. Four database search engines were used (Comet [69], Xtandem [70], MSGF [71] and Myrimatch [72], with search parameters as above. Each search was followed by the application of the PeptideProphet [73] tool; the iprophet [74] tool was then used to combine the search results, which were filtered at the false discovery rate of 0.01; furthermore, the identification was only accepted if at least two of the search engines agreed on the identification. Protein inference was performed with ProteinProphet [75]. For those protein groups accepted by a false discovery rate filter of 0.01, a Normalized Spectral Abundance Factor (NSAF) [76] was calculated based on the peptide to spectrum match count; shared peptides were accounted for by the method of [77], giving normalised spectral abundance factor (dNSAF) values. The dNSAF values were used to calculate abundance as a % of the UPF1 bait protein, which were reported for the NMD factors and SFV proteins that were detected in the samples.

## Small-scale SFV protein expression and puromycin incorporation assays

HeLa cells ($1.5 \times 10^5$ per well) were seeded into 6-well plates. The following day, 1.25 μg of the relevant pcDNA5.3xFLAG plasmids were transfected using Dogtor transfection reagent (OZ Biosciences), as per manufacturer's recommendations. Cells were harvested 48 hours later, after which $1 \times 10^6$ cells per condition were collected to make lysates for western blot analysis, while the remaining cells were collected for RNA analysis. The cells were harvested at 4°C for 5 minutes at 250 x g, washed once with 1x PBS, and resuspended in either 2x SDS-PAGE Loading Buffer (for protein analysis) or 900 μL of TRI reagent (for RNA analysis). For the puromycin incorporation assays, cells were transfected as above and prior to harvesting, the medium was aspirated and medium containing either DMSO or 100 μg/mL cycloheximide (CHX) was added to the cells and incubated for 2 hours at 37°C. Thereafter, the cells were pulse labelled for 10 minutes with puromycin (10 μg/mL). After the 10 minutes pulse, as a recovery step, medium containing either DMSO or CHX was re-added to the cells for 30 minutes at 37°C. Cells were harvested as above and $1 \times 10^6$ cells per condition were resuspended in 2x SDS-PAGE loading buffer and incubated at 95°C for 5 minutes for western blot analyses.

## Western blot analysis

Lysates were loaded into either 10% SDS-PAGE gels or pre-casted 4–12% Bis-Tris 26-well NuPAGE or 10-well Bolt gels (Invitrogen, Thermo Fisher Scientific) and electrophoresed in either 1x SDS-PAGE running buffer or 1x MOPS running buffer. Proteins were transferred onto nitrocellulose membranes using either the BioRAD Semi-Dry Blot system (30 minutes in 1xBjerrum buffer + 20% methanol) for SDS-PAGE gels or the iBlot2 (P0, 7mins, using iBlot 2NC regular stacks) (Invitrogen, Thermo Fisher Scientific) for pre-casted gels. Membranes were blocked for 1 hour at room temperature (RT) in 5% milk-TBS-T (0.1% Tween20), or in the case of Rb anti-eIF2α, Ms anti-eIF2α and Rb anti-p-eIF2α blots, 5% milk-TBS, and then incubated with the indicated primary antibodies at 4°C overnight. Primary antibodies were diluted in 5% milk-TBS-T, apart from Rb anti-eIF2α, Ms anti-eIF2α and Rb anti-p-eIF2α, which were diluted in 5% BSA-TBS-T (0.1% Tween20). After three 10 minutes washes in TBS-T, membranes were incubated with the indicated secondary antibodies in 5% milk-TBS-T for 1.5 hours at RT, followed by three 10 minutes TBS-T washes. The blots were then visualised using the Odyssey System (LICOR).

## Polysome fractionations

The protocol for polysome fractionations was adapted from [78]. HeLa cells were transfected with the 'empty' (pcDNA5_FRT_TO_3xFlag) or 'capsid' (pcDNA5.3xFLAG-capsid) plasmids as described for the large scale transfections above. The cells were treated with 100 μg/mL CHX for 4 mins at 37°C prior to harvesting. Cells were washed once with ice-cold PBS containing 100 μg/mL CHX, scraped off the surface of the dish with 750 μL per 15cm dish of PBS-CHX and then transferred into tubes. Cells were collected by centrifugation at 4°C for 5 minutes at 500 x *g* and lysed in 600 μL of lysis buffer (10 mM Tris-HCl [pH 7.5], 10 mM NaCl, 10 mM MgCl$_2$, 1% Triton X-100, 1% sodium deoxycholate, supplemented fresh with 1 mM DTT, 0.2 U/μL RNase Inhibitor [NxGen] and 100 μg/mL CHX) on ice with occasional vortexing. Thereafter, cell lysates were cleared by centrifugation at 4°C for 5 minutes at 16000 x g and loaded onto ice-cold 15–50% sucrose-CHX gradients and ultracentrifuged at 4°C for 2 hours at 274 000 x *g* in a SW-41Ti rotor (Beckman). Gradients were fractionated and monitored at an absorbance of 254 nm with a fraction collector (BioComp Instruments) at a speed of 0.2-mm/s, with a distance of 3.71 mm per fraction.

### Subcellular fractionations and rRNA analysis

HeLa cells ($2.5\times10^6$ per dish) were seeded into 15 cm dishes and transfected with the relevant pcDNA5.3xFLAG plasmids as described previously. Cells were harvested 48 hours later: $1\times10^7$ cells per condition 'empty' or 'capsid' were collected and fractionated into nuclear and cyto-plasmic fractions according to [79]. For RNA analysis, 900 μL of TRI-Reagent was added to $1\times10^6$ cell equivalents of each fraction (Cyto and nuclear) and RNA extracted as described above. The quality of the 18S and 28S rRNA in the fractions was assessed using the 2100 Bioa-nalyzer Instrument (Agilent) with Agilent RNA 6000 Nano Kit, as per the manufacturer's instructions. The 28S rRNA and 45S pre-rRNA of the nuclear fractions were further analysed by RT-qPCR using the following primers: 28S rRNA: fwd: 5'-AGAGGTAAACGGGTGGGG TC-3'; rev: 5'-GGGGTCGGGAGGAACGG-3' and 45S pre-rRNA: fwd: 5'-GAACGGTGGTG TGTCGTT-3'; rev: 5'-GCGTCTCGTCTCGTCTCACT-3'.

## Supporting information

**S1 Fig. Coomassie-stained gels showing eluate samples of SFV affinity purifications that were sent for mass spectrometry analysis.** Geldoc images of coomassie-stained gels showing eluates (20 μL) from three biological replicates of each SFV affinity purification electropho-resed 1 cm into the gels, prior to cutting. Rectangular segments (10 mm x 3 mm) for each lane were cut from the gel and samples were processed for mass spectrometry analysis. Note that samples of Rep1: nsp3-Z were loaded in opposite positions, hence highlighted in red and denoted (+) and (-).
(PDF)

**S2 Fig. Anti-flag western blots of SFV affinity purifications with (bottom) and without (top) treatment with RNaseA.** Red asterisks indicate 3xFLAG tagged SFV proteins (or the 3xFLAG alone) at respective sizes. 'Untr' refers to an untransfected control condition that underwent the affinity purification procedure, while 'empty' denotes transfection with a plas-mid construct containing only the 3xFLAG tag (with no additional coding region) followed by affinity purification. The expected sizes of the proteins (3xFLAG included) were: empty ~8kDa; nsp1 ~63kDa; nsp2 ~92kDa; nsp3-Z ~82kDa; nsp4 ~72kDa; capsid ~33kDa and Env ~111kDa. The left side of the WBs indicate the Eluates (after flag peptide elution from the beads) of each SFV affinity purification, which were sent for mass spectrometry analysis. The right hand side of the WBs indicate the SFV proteins that were still present on the beads after the flag peptide elution step.
(PDF)

**S3 Fig. A network visualisation of the SFV-ribosomal host protein interactome.** In total, 77 ribosomal host proteins are displayed. SFV proteins are depicted in black circles and host proteins are displayed in smaller colour-coded circles according to the key. SFV-host PPI are depicted with solid grey lines. Grey boxes reflect host proteins that were identified as interactors to more than one of the SFV proteins. In these cases, the solid grey lines connect the grouped set of host pro-teins to the SFV proteins for which they were identified as interactors. Host-host PPI ascertained through STRING analysis are depicted with dotted pink lines. The host interactors were collected based on three independent biological replicates. SFV-ribosomal host protein interactors identi-fied through a genome-wide fluorescence microscopy based siRNA screen as exhibiting potential proviral roles in SFV replication are additionally shown (turquoise borders, II $\leq$ 0.5). None were identified as having a potential antiviral role (Threshold $\geq$ 1.3).
(PDF)

**S4 Fig. (Related to Fig 5). SFV infection inhibits NMD at 3–4 hours post infection.** RT-qPCR analysis showing mRNA levels relative to Actin-β of the viral RNA (first graph–'ZSG'), NMD targets (black lines) and non-NMD targets (light grey lines) normalised to either the 0.5 hr sample (in the case of the viral RNA 'ZSG') or the Uninfected control, at different time points post SFV-ZSG infection. Cells were infected with an MOI of 10. Line graphs display the mean ± SD of four independent biological replicates (n = 4). Statistical significance was determined using unpaired, two-tailed t-tests, corrected for multiple comparisons using the Holm-Šídák method, with alpha = 0.05 (GraphPad Prism, v8.4.1). Asterisks indicate adjusted p values: $p \leq 0.05^{*}$, $p \leq 0.01^{**}$, $p \leq 0.001^{***}$.
(PDF)

**S5 Fig. (Related to Fig 6). SFV capsid expression did not alter polysome profiles or rRNA. a,** HeLa cells expressing 'empty' (pcDNA5_FRT_TO_3xFlag) or 'capsid' (pcDNA5.3xFLAG-capsid) plasmids were treated with CHX [100 μg/mL], lysed and ultracentrifuged (274 000 x *g*, 2 hours, 4˚C) through a 15–50% sucrose-CHX gradient to separate ribosomal subunits and monosomes from active/translating polysomes (Protocol adapted from [78]). The gradients were fractionated and the absorbance values (254 nm) throughout the gradient is plotted. **b,** Bioanalyzer analyses of RNA extracted from cytoplasmic (CYTO) and nuclear (NUCL) fractions of 'empty' or 'capsid' expressing cells. The peaks at ~42 and 49 [s] indicate the 18S and 28S rRNAs, respectively. RIN values indicate RNA integrity, 10 being the highest. **c,** RT-qPCR analysis showing raw CT values of 28S and 45S rRNAs from 'empty' or 'capsid' expressing cells.
(PDF)

**S1 Table. Interactors identified by mass spectrometry.** Excel file with mass spec datasets of the immunoprecipitates of the 3xFLAG-tagged nsP1, nsP2, nsP3-Z, nsP4, Envelope and Capsid proteins.
(XLSX)

**S2 Table. SiRNA screen data.** Excel file with raw data of the genome-wide siRNA screen performed to identify host cell proteins that inhibit (antiviral) or enhance (proviral) SFV replication.
(XLSX)

**S3 Table. Data of Gene Ontology (GO) enrichment analysis for the MCODEs shown in Fig 4.**
(XLSX)

## Acknowledgments

We are grateful to Sophie Braga Lagache and Manfred Heller of the Proteomics Mass Spectrometry Core Facility (PMSCF) of the University of Bern for mass spectrometry sample preparations and measurements. pcDNA5_FRT_TO_3xFlag(N) was a kind gift from Christian Kroun Damgaard (Aarhus University).

## Author Contributions

**Conceptualization:** Lara Contu, Oliver Mühlemann.

**Data curation:** Lara Contu, Giuseppe Balistreri, Anne-Christine Uldry.

**Formal analysis:** Lara Contu, Giuseppe Balistreri, Michal Domanski, Anne-Christine Uldry.

**Funding acquisition:** Oliver Mühlemann.

**Investigation:** Lara Contu, Giuseppe Balistreri, Michal Domanski, Oliver Mühlemann.

**Methodology:** Lara Contu, Giuseppe Balistreri, Michal Domanski, Anne-Christine Uldry.

**Project administration:** Lara Contu, Giuseppe Balistreri, Oliver Mühlemann.

**Resources:** Michal Domanski.

**Software:** Lara Contu, Anne-Christine Uldry.

**Supervision:** Michal Domanski, Oliver Mühlemann.

**Validation:** Lara Contu.

**Visualization:** Lara Contu.

**Writing – original draft:** Lara Contu, Oliver Mühlemann.

**Writing – review & editing:** Lara Contu, Giuseppe Balistreri, Michal Domanski, Anne-Christine Uldry, Oliver Mühlemann.

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
