## [Decision Letter · Decision Letter 0]

9 Dec 2020

Dear Dr. Mühlemann,

Thank you very much for submitting your manuscript "Characterisation of the Semliki Forest Virus-host cell interactome reveals the viral capsid protein as an inhibitor of nonsense-mediated mRNA decay" for consideration at PLOS Pathogens. As with all papers reviewed by the journal, your manuscript was reviewed by members of the editorial board and by several independent reviewers. The reviewers appreciated the attention to an important topic. Based on the reviews, we are likely to accept this manuscript for publication, providing that you modify the manuscript according to the review recommendations.

The manuscript was reviewed by individuals with expertise in alphaviruses and/or host-pathogen interactions. Overall, all three reviewers were quite positive in their assessment of the work, commenting favorably on the nature of the experimental approach and the significance of the findings. However, reviewers did have a number of questions and suggestions. In particular, (i) Reviewer 1 recommended additional data analysis and commentary, while Reviewer 2 and 3 recommended that select additional experiments be considered. Although these are generally minor issues, they will need to be thoroughly addressed and the manuscript appropriately modified before being further consideration for publication. Thank you again for submitting such an outstanding manuscript to the journal. We look forward to receiving the revised manuscript in the near future and its ultimate publication.

Sincerely,

John T. Patton, PhD

Associate Editor

PLOS Pathogens

Mark Heise

Section Editor

PLOS Pathogens

Kasturi Haldar

Editor-in-Chief

PLOS Pathogens

orcid.org/0000-0001-5065-158X

Michael Malim

Editor-in-Chief

PLOS Pathogens

orcid.org/0000-0002-7699-2064

The manuscript was reviewed by individuals with expertise in alphaviruses and/or host-pathogen interactions. Overall, all three reviewers were quite positive in their assessment of the work, commenting favorably on the nature of the experimental approach and the significance of the findings. However, reviewers did have a number of questions and suggestions. In particular, (i) Reviewer 1 recommended additional data analysis and commentary, while Reviewer 2 and 3 recommended that select additional experiments be considered. Although these are generally minor issues, they will need to be thoroughly addressed and the manuscript appropriately modified before being further consideration for publication. Thank you again for submitting such an outstanding manuscript to the journal. We look forward to receiving the revised manuscript in the near future and its ultimate publication.

Reviewer Comments (if any, and for reference):

Reviewer's Responses to Questions

**Part I - Summary**

Reviewer #1: The authors cataloged host proteins that bind to each of the expressed SFV proteins, identified host proteins showing pro- and anti-viral functions by using siRNA screening, showed SFV replication-induced inhibition of the NMD pathway late in infection and capsid-induced inhibition of the NMD pathway.

Although there is a possibility that all of the detected interactions may not occur in infected cells, their data will serve as a foundation for further understanding of interactions between host proteins and viral proteins in alphavirus-infected cells.

The biological significance of the capsid-induced inhibition of the NMD pathway on SFV replication was not experimentally explored. Does expression of capsid protect SFV RNA from NMD or rapid decay? Inclusion of data addressing this question would strengthen the significance of the present study.

Reviewer #2: The manuscript from Contu et al. describes the characterization of the Semliki Forest Virus (SFV)-host interactome, and reveals a role for SFV capsid protein in the suppression of nonsense mediated decay (NMD). The authors utilize over-expression of individual SFV proteins followed by affinity purification to isolate viral-host protein complexes. Composition of these complexes was determined by quantitative mass spectrometry to identify interactors. Inclusion of RNase in sample preparation allowed the determination of those interaction that were dependent on RNA. Interaction networks were described on the basis of viral interacting partner (Fig 3) and on the basis of host cell gene ontology descriptions (Fig 4). The authors found numerous host proteins that interacted with multiple viral proteins, and enrichment of interacting partners in particular cellular functions, of note ribosome biogenesis, RNA binding and processing. Some of these had not been found in previous studies, but a listing of newly identified interactors did not appear in a figure. In parallel to the proteomic analysis the investigators performed a genome wide siRNA screen utilizing a SFV expressing a reporter GFP to assess both pro- and anti-viral effects. A proportion of interactors were found to have an impact on viral replication. Components of the NMD pathway were found to have an antiviral function which is consistent with previous studies from this group showing SFV to be susceptible to restriction by NMD and functionally inhibit NMD to facilitate efficient replication. Interaction with NMD was determined for each individual viral protein through the examination of known cellular NMD target transcripts. Expression of capsid protein increased the abundance of NMD target transcripts while not altering the abundance of control transcripts.

Overall the manuscript is clearly written, the mass spec analyses seem rigorous, as does the interactor analysis. The datasets will be of use to the broader field, and the finding that capsid inhibits NMD is significant.

Reviewer #3: There is a growing understanding that viruses use diverse strategies to control the cellular translation and RNA decay machinery. In this manuscript, Contu and colleagues use proteomics to investigate host proteins that negatively or positively regulate Semliki Forest virus replication. Comparison of spectrometry of proteins associated with transiently expressed viral proteins to results of an RNAi screen identified many proteins involved in translation, ribosome biogenesis, and other aspects of RNA metabolism. Of these, the authors focus on UPF1 and the process of nonsense-mediated decay, showing that viral infection inhibits translation and NMD with similar kinetics. In addition to the likely indirect effect of translation inhibition on NMD, they also find that overexpression of capsid, but not several other SFV proteins, causes increased expression of several NMD target mRNAs, without an apparent effect on translation. The experiments are carefully performed, and the manuscript is well written. The apparently specific, translation-independent effect of capsid protein on NMD is of particular interest, although more insight into the mechanism of this inhibition would be welcome.

**Part II – Major Issues: Key Experiments Required for Acceptance**

Reviewer #1: The authors stated that interactions of nsP3-Z and capsid with UPF1 is RNA-dependent. As experiments testing interactions of UPF1 with nsP3-Z and capsid shown in Fig. 6d were done without RNase treatment, this statement lacks solid experimental data. The authors should perform co-immunoprecipitation analysis shown in Fig. 6d by using cell extracts that are treated with RNase.

Reviewer #2: no additional experiments, however some further data analysis:

1. The authors reference having confirmed interactions found in previous studies, and having found new interactors. It would be helpful to specify how many of the newly identified interactors have pro- or antiviral function. My main concern is that, in contrast to similar previous studies, the authors are expressing individual viral protein rather than examining the interactions in the context of an infection. While I believe the data generated to be of value, this approach does lend itself to the possibility of aberrant interactions that are not relevant in the context of an infection. Presentation of the number of these new, previously unrecognized, interactors with an impact on viral replication would counter this issue.

2. The authors should address in some way why only a small handful of the interactors have an impact on virus replication. In looking at the siRNA spreadsheet it appears that there are over 3000 of the 38,000 knockdowns that meet the threshold for pro- or antiviral. Of the 251 interactors it appears that 30 (?) have an impact on viral replication. Is this level of enrichment of factors impacting replication in a set of interacting proteins what would be expected? The proportion of interactors impacting viral replication should be clearly stated. An explanation of why more identified interactors do not impact replication would be helpful. The low number of interactors influencing virus becomes very obvious in figure 4 where the interaction nodes shown have very few factor with pro- or antiviral effects, this should be addressed.

Reviewer #3: 1. The major drawback of the study is a lack of insight into why the capsid protein interferes with turnover of NMD target mRNAs. The authors look for changes in polysome profiles andrRNA processing, but have they instead s investigated whether capsid overexpression interferes with interactions between UPF1 and other proteins involved in NMD or NMD target RNAs?

2. Conversely, have the authors tried to use ISRIB or a PKR-deficient cell line to test whether there is evidence that viral infection inhibits NMD independent of translational repression? Further evidence for the physiological role of capsid-mediated NMD inhibition outside of the context of individual viral protein expression would be very helpful to establish the significance of the findings.

3. Because of the central importance of establishing that capsid overepxression does not inhibit translation, quantification of the puromycylation assay in Figure 6d should be provided.

**Part III – Minor Issues: Editorial and Data Presentation Modifications**

Reviewer #1: 1. Experiments shown in Fig. 1 included treatment of samples with RNase A. How did the authors know that the RNase treatment condition was appropriate? They should include data demonstrating that the experimental approach for RNase A treatment (page 23) were appropriate for RNA degradation.

2. They claimed that expressed SFV proteins, including capsid and nsP3-Z, did not suppress translation nor induce eIF2a phosphorylation (Fig. 6), and concluded that capsid-induced NMD inhibition was not due to translational suppression. This conclusion was based on the assumption that capsid was expressed in most of the cells, yet this assumption was not experimentally tested. The current data do not exclude the possibility that only low levels of cells expressed capsid, which might have suppressed translation, leading to NMD inhibition. To exclude this possibility, they should show the percentages of cells expressing each of the viral proteins.

3. Page 4. Thy used nsp3 carrying ZSG tag for protein expression, whereas no explanation about this tag was given at the beginning section of “Results”. The authors should explain this tag and a rationale for the insertion of this tag within the nsp3 gene in first paragraph or early section of “Results” section.

Reviewer #2: Some justification for choosing 1.3 and 0.5 as the II cutoffs is needed. Is this just arbitrary or is there a reason these values were chosen?

While the authors focused on the interactors in the RNA containing samples, it is hard to know what to make of the changes in interactors in samples treated with RNase. The samples have no viral RNA present so RNA dependent interactions are not viral RNA dependent interactions, this should be clearly stated. Also, should any relevance be placed in the difference in interaction in the absence of RNA? This seems a very artificial situation.

Page 6 , lines 6-9: Why does the fact that heat maps and silver stained gels are consistent provide confidence in the approach to sample acquisition? These are just two different analyses of the same samples. Surely this provides confidence in the two techniques for analysis as they are consistent with one another.

How do siRNA knockdowns impact cell viability? Is there some inherent within the analysis that accounts for siRNA induced cell death?

Page 14, line 22 and Fig 5d: nsP2 was shown through MS to interact with UPF-1 but does not in 5d, conversely nsP1 was not seen to interact with UPF-1 by MS but does in 5d. Explanation?

Page 16, line 14-15: There’s no reason not to show data, please add it.

Page 25-26: For the MCODE method explanation Fig 5 is referenced and I believe this should be Fig 4.

Reviewer #3: 1. Figure 4: The authors provide direct evidence of NMD compromise, but I think the GO analysis is somewhat misleading. The NMD go-term contains all of the ribosomal proteins, which means that any experiment in which ribosomes are enriched will lead to apparent enrichment for "NMD". This is compounded by the fact that for the purposes of GO analysis, ribosomal proteins are treated as independent entities, despite the fact that they are often obligate members of pre-ribosomes/ribosomes. From the supplemental table, it appears that only UPF1 among recognized NMD proteins (along with PABPC1, an inhibitor of NMD) contributes to this classification.

2.Page 16, last sentence of discussion: It's not clear what is meant by the sentence, "Though capsid could be trapped in the nucleus upon blocking of export…"

PLOS authors have the option to publish the peer review history of their article (what does this mean?). If published, this will include your full peer review and any attached files.

Reviewer #1: No

Reviewer #2: No

Reviewer #3: No
---

## [Editor Report · Decision Letter 1]

3 May 2021

Dear Dr. Mühlemann,

We are pleased to inform you that your manuscript 'Characterisation of the Semliki Forest Virus-host cell interactome reveals the viral capsid protein as an inhibitor of nonsense-mediated mRNA decay' has been provisionally accepted for publication in PLOS Pathogens.

Best regards,

John T. Patton, PhD

Associate Editor

PLOS Pathogens

Mark Heise

Section Editor

PLOS Pathogens

Kasturi Haldar

Editor-in-Chief

PLOS Pathogens

orcid.org/0000-0001-5065-158X

Michael Malim

Editor-in-Chief

PLOS Pathogens

orcid.org/0000-0002-7699-2064

Thank you very much for submitting your revised manuscript entitled "Characterisation of the Semliki Forest Virus-host cell interactome reveals the viral capsid protein as an inhibitor of nonsense-mediated mRNA decay" (PPATHOGENS-D-20-02414R1) for review by PLOS Pathogens. The modifications made to the manuscript and outstanding efforts in responding to the reviewer's comments and concerns are much appreciated. I am pleased to accept the manuscript for publication.
---

## [Editor Report · Acceptance letter]

18 May 2021

Dear Dr. Mühlemann,

We are delighted to inform you that your manuscript, "Characterisation of the Semliki Forest Virus-host cell interactome reveals the viral capsid protein as an inhibitor of nonsense-mediated mRNA decay," has been formally accepted for publication in PLOS Pathogens.

Best regards,

Kasturi Haldar

Editor-in-Chief

PLOS Pathogens

orcid.org/0000-0001-5065-158X

Michael Malim

Editor-in-Chief

PLOS Pathogens

orcid.org/0000-0002-7699-2064